# Climate change impacts model parameter sensitivity - implications for calibration strategy and model diagnostic evaluation

Lieke Anna Melsen[1] and Björn Guse[2,3]

[1]Hydrology and Quantitative Water Management, Wageningen University, Wageningen, the Netherlands
[2]GFZ German Research Centre for Geosciences, Section Hydrology, Potsdam, Germany
[3]Christian-Albrechts-University of Kiel, Institute of Natural Resource Conservation, Department of Hydrology and Water Resources Management, Kiel, Germany

**Correspondence:** Lieke Melsen (lieke.melsen@wur.nl)

**Abstract.** Hydrological models are useful tools to explore the impact of climate change. To prioritize parameters for calibration and to evaluate hydrological model functioning, sensitivity analysis can be conducted. Parameter sensitivity, however, varies over climate, and therefore climate change could influence parameter sensitivity. In this study we explore the change in parameter sensitivity for the mean discharge and the timing of the discharge, within a plausible climate change rate. We investigate if changes in sensitivity propagate into the calibration strategy, and diagnostically compare three hydrological models based on the sensitivity results. We employed three frequently used hydrological models (SAC, VIC, and HBV), and explored parameter sensitivity changes across 605 catchments in the United States by comparing GCM(RCP8.5)-forced historical and future periods. Consistent among all hydrological models and both for the mean discharge and the timing of the discharge, is that the sensitivity of snow parameters decreases in the future. Which other parameters increase in sensitivity is less consistent among the hydrological models. In 45% to 55% of the catchments, dependent on the hydrological model, at least one parameter changes in the future in the top-5 most sensitive parameters for mean discharge. For the timing, this varies between 40% and 88%. This requires an adapted calibration strategy for long-term projections, for which we provide several suggestions. The disagreement among the models on the processes that become more relevant in future projections also calls for a strict evaluation of the adequacy of the model structure for long-term simulations.

## 1  Introduction

Earth and environmental computer models are indispensable tools to explore an uncertain future. Whereas observational studies report on historical changes in streamflow patterns across the contiguous United States (CONUS) that might be attributed to climate change (Stewart et al., 2005; Sagarika et al., 2014), hydrological models are applied in the same region to gain insights into long term changes in the future (e.g. Mizukami et al., 2016; Melsen et al., 2018; Chegwidden et al., 2019; Brunner et al., 2020). These model projections can support water resource managers to prepare for future changes. The models are thus related to costly and impactful decisions (McMillan et al., 2017; Metin et al., 2018).

Given the relevant role of models to support decision making, model functioning should be thoroughly scrutinized. A frequently used tool to evaluate hydrological model functioning is sensitivity analysis (Pianosi et al., 2016; Devak and Dhanya, 2017). Sensitivity analysis is aimed at identifying the relative impact of model parameters on model response. The results of a sensitivity analysis differs over models, target variable of the model response, and applied sensitivity analysis methods (Shin et al., 2013; Razavi and Gupta, 2015; Guse et al., 2016a; Haghnegahdar et al., 2017; Mai and Tolson, 2019).

However, parameter sensitivity also differs across climate, as for instance showed by Demaria et al. (2007), Van Werkhoven et al. (2008) and Melsen and Guse (2019): In a cold catchment with a large fraction of the precipitation falling as snow, snow parameters are supposed to be sensitive, while in a tropical catchment without snowfall, snow parameters are not supposed to show any sensitivity. As such, it is common understanding that parameter sensitivity depends on climate. But reconsidering this fact, this would also imply that parameter sensitivity could change in a changing climate. Therefore, the question is whether, within a plausible rate of climate change, hydrological parameter sensitivity changes. This could have consequences for the way hydrological models should be calibrated for long term projections. Besides, it offers the opportunity to compare different models to evaluate if the same mechanisms are simulated as being relevant for future changes.

Many hydrological models that are used for long term projections have parameters that require calibration to identify their values for the catchment under study. Hydrological model parameters are generally calibrated on discharge time series. But discharge is a lumped catchment response, and therefore only provides limited catchment information (Jakeman and Hornberger, 1993; Guse et al., 2016a). A rule of thumb suggested by Beven (1989) and employed by many modelers is that, given the limited information available in a discharge time series, three to five parameters can be calibrated based on these data (and this might already be a high number, following the famous quote of von Neumann[1]). Global Sensitivity Analysis (GSA) can be employed to identify the parameters that have most influence on the model output (Demaria et al., 2007; van Werkhoven et al., 2009; Pianosi et al., 2016; Borgonovo et al., 2017; Zadeh et al., 2017). Subsequently, the three to five parameters that show most sensitivity are selected for calibration. However, if parameter sensitivity changes with climate change, this could interfere with the parameter prioritization procedure for models used for long term projections.

Besides the potential consequence for calibration, evaluating the relation between change in parameter sensitivity and climate change also provides the opportunity to diagnostically evaluate model functioning during long-term projections. Several studies already investigated the change in parameter sensitivity over time, focusing on specific events or relatively short time scales (Reusser et al., 2011; Herman et al., 2013b; Guse et al., 2014; Massmann and Holzmann, 2015; Pfannerstill et al., 2015). For instance, it was demonstrated that certain parameters are triggered during specific discharge conditions such as high or low flow events (Pfannerstill et al., 2015; Guse et al., 2016b). If certain events will prevail or become less frequent in a future climate, this might change the average parameter sensitivity over the long run. As such, evaluating long-term changes in parameter sensitivity can provide insights into systemic changes. By comparing changes among several model structures, the robustness

---

[1] *"With four parameters I can fit an elephant, with five I can make him wiggle his trunk"*, John von Neumann (1903-1957)

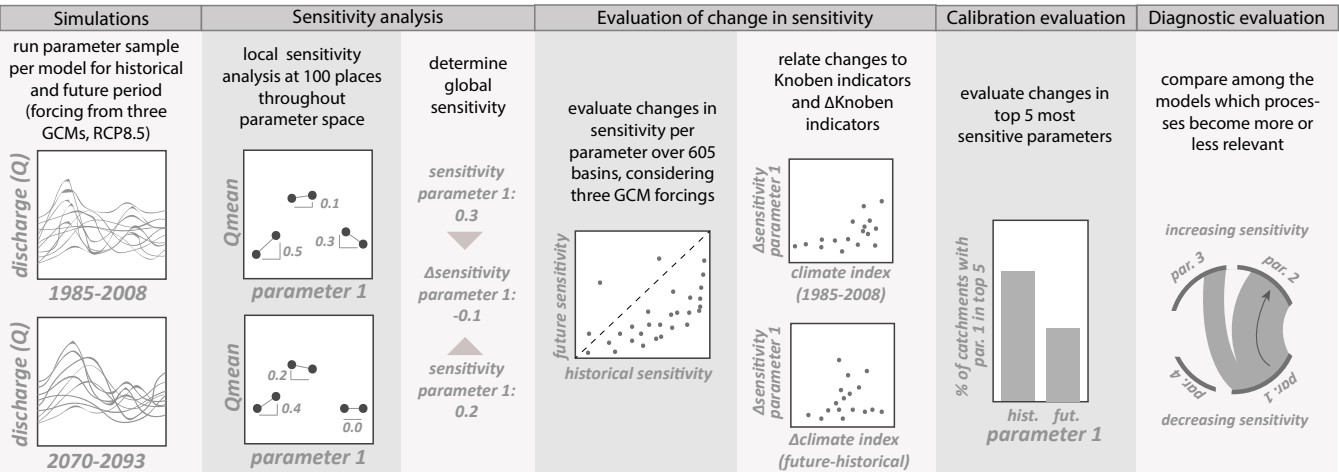

**Figure 1.** Summary of the methodological approach, to be read from left to right. The first three panels are the calculations, the other four panels are the actual analyses.

of simulated systemic changes can be evaluated.

In this study we investigate how parameter sensitivity changes as a consequence of climate change. We evaluate if and how this has consequences for parameter prioritization for calibration, and if systemic changes are robust across different hydrological model structures. To this end, we apply a hybrid local-global sensitivity analysis method to three frequently used hydrological models in 605 basins across the US. We evaluate two signatures (mean discharge and timing of the discharge), and link changes in sensitivity to changes in climate. To sample plausible climate change space, we employ forcings from three different GCMs. Finally, we evaluate the impact on the top-5 most sensitive parameters in each basin, and investigate the transmission of sensitivity from one parameter to the other. We end with a recommendation on how to account for changes in sensitivity in the calibration strategy of models used for long-term projections, and an evaluation of the robustness of systemic changes across different models.

## 2   Methods

To investigate changes in parameter sensitivity, we employed three frequently used hydrological models. The models were run for a historical and a future period over 605 catchments, forced with three bias corrected and statistically downscaled global circulation model. A hybrid local-global sensitivity analysis method was applied to the simulations of both periods, evaluating two target variables; the mean discharge, and the day of the year on which half of the yearly discharge has passed (discharge timing). Then, the differences in parameter sensitivity between the historical and future period were explored in several ways.

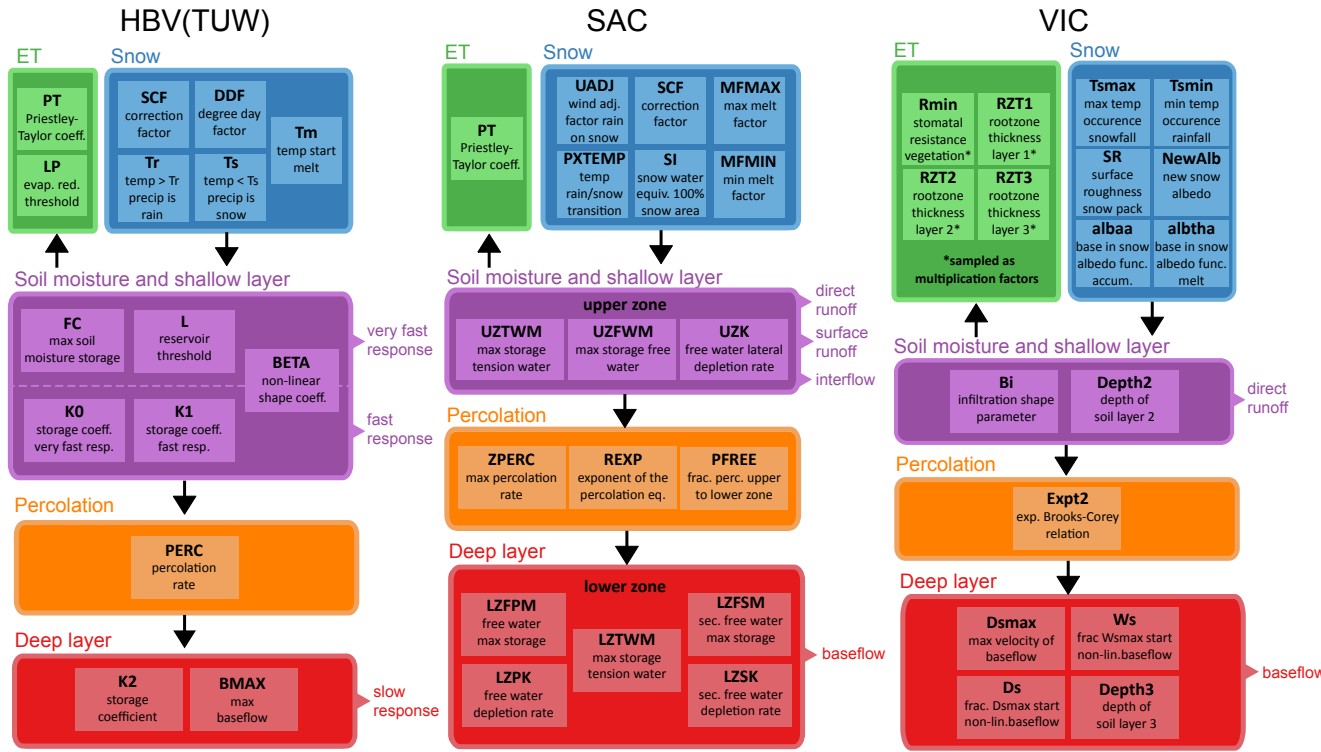

**Figure 2.** Simplified representation of the model structure of the three models employed in this study. All the parameters that are displayed are included in the sensitivity analysis. Parameters are colored according to the flux or state they influence (evapotranspiration (ET), snow, soil moisture and shallow layer, percolation, deep layer). The colors are used consistently throughout all the figures in this study. Parameter boundaries can be found in Appendix Tables A1, A2, and A3.

First, per parameter to investigate which parameters change, and over different climate indicators to investigate how climate and climate change explain changes in sensitivity. Then, we assessed how the top-5 most sensitive parameters would change in the future period, thereby impacting the calibration strategy. Finally, we conduct a diagnostic model evaluation, amongst others by investigating the transmission of sensitivity from one parameter to the other. An overview of the procedure is shown in Figure 1.

## 2.1 Models

We investigated for three models whether parameter sensitivity changes within a plausible climate change range: the TUW-model following the structure of HBV (Parajka et al., 2007, hereafter referred to as HBV), SAC-SMA combined with SNOW-17 (Newman et al., 2015), and VIC 4.1.2h (Liang et al., 1994). All three models have previously been used for long-term climate impact projections: e.g. Teutschbein et al. (2011); Wetterhall et al. (2011) for HBV; Koutroulis et al. (2013); Peleg et al. (2015)

for SAC-SMA; and Christensen et al. (2004); Wu et al. (2012) for VIC, and are therefore relevant models to consider. The same suit of models was explored in another context in Melsen et al. (2018) and Melsen and Guse (2019).

A simplified representation of the model structures, including a description of the parameters that were accounted for in the sensitivity analysis, are displayed in Figure 2. A more elaborate description of each model can be found in Melsen et al. (2018),
and in the respective references of each models: Bergström (1976, 1992) for HBV, Burnash et al. (1973); National Weather Service (2002) for SAC-SMA, and Liang et al. (1994, 1996) for VIC.

All three models have a different structure, but HBV and SAC are more alike in terms of structure and conceptualization than VIC. A difference is that for HBV and SAC, potential evapotranspiration was obtained with the Priestley-Taylor equation
(Lhomme, 1997), while for VIC, the Penman-Monteith equation was employed. Also, SAC and HBV contain conceptual snow modules based on a degree-day approach, while VIC has an energy-balance-based snow implementation.

## 2.2  Catchments and Forcing

All three models were run for a historical and future period of 28 years, of which the first five years were omitted from both periods for spin up. As such, the historical period that is analyzed covers 1985-2008, and the future period 2070-2093, 23 years
each. The forcing for both periods was obtained from statistically downscaled and bias corrected output from three GCMs: the Max Planck Institute for Meteorology Earth System Model MR (MPI-ESM-MR, Giorgetta, 2013), the Community Climate System Model 4.0 (CCSM4, Gent et al., 2011) and the The Institut Pierre Simon Laplace model (IPSL-CM5A-MR, Dufresne, 2013). All GCMs are from the Climate Model Intercomparison 5 (CMIP5), using Representative Concentration Pathway 8.5 (RCP8.5). Bias correction was done according to the Bias Correction and Spatial Disaggregation (BCSD) method of Wood
et al. (2004), based on the Maurer et al. (2002) forcing data.

Our study is an investigation of the potential that a plausible climate change rate might impact hydrological model parameter sensitivity. As such, we only selected a subset of three GCMs to sample plausible climate change rates. The three selected GCMs represent different climate model families as identified by Knutti et al. (2013), to capture the spread among the differ-
ent GCMs. Within each family, the selected models are among the better performing ones when evaluated against observed precipitation and temperature (Sheffield et al., 2013).

Finally, the GCM forcing was lumped over the CAMELS basins. The CAMELS data set contains forcing, discharge observations, and catchment characteristics for 671 catchments throughout the contiguous United States with limited human impact
(Newman et al., 2014, 2015; Addor et al., 2017). We employed a subset of 605 catchment-averaged forcings, because at the time of calculation, there were still issues with determining the exact catchment area for the remaining 66 catchments.

The hydrological models were not calibrated, since we employed global sensitivity analysis across the full parameter range. Therefore, the 605 simulated catchments should be perceived as 605 different climate instances with an individual level of climate change, rather than as catchment representative models. Given that each catchment was forced with the three GCMs means that in total, 605·3=1815 different evaluations of changes in sensitivity were conducted. The models are, however, able to achieve acceptable model performance in these basins when forced with observations and confronted with discharge observations (Melsen et al., 2018), providing credibility to the models to be used in this context.

The goal of this study is to investigate how climate might impact parameter sensitivity within a plausible climate change range. We are thus particularly interested in to which extent we can expect changes in the sensitivities of model parameters as a consequence of changes in the climate. As such, it is of second order importance whether the climate model gives highly accurate predictions or whether the hydrological model can exactly capture catchment behavior. It is, however, important to note that we employed the highest emission scenario (RCP8.5), thereby investigating the effect of the higher ranges of plausible climate change. It can be expected that the impact of climate change on parameter sensitivity will be lower for lower emission scenarios. However, RCP 8.5 is often used to provide an upper boundary for long-term projections, thereby demonstrating the relevance of choosing this scenario.

## 2.3 Sensitivity analysis methodology

In the selection of the sensitivity analysis method, a few points were considered. First, it had to be a global method, because global sensitivity analysis methods are used to identify the most sensitive parameters for calibration (whereas local methods are generally applied after calibration). Secondly, we had to account for a high number of runs (605 basins, three GCMs, two periods, three hydrological models). Therefore, we selected the hybrid local-global method DELSA (Rakovec et al., 2014), which is computationally cheaper than traditional (variance-based) global sensitivity analysis methods.

DELSA evaluates local sensitivity at several places throughout parameter space, as such mimicking global sensitivity analysis. First, 100 parameter samples called base-runs were created based on a space filling sampling strategy. The models were run for all 100 samples. Secondly, the parameters are one-at-a-time perturbed with 1% compared to their base-run value. The effect of this perturbation on the model output, compared to the corresponding base-run, represents the parameter sensitivity:

$$\frac{\partial \psi}{\partial \theta_k}\bigg|_{\theta_{base-run_i}} = \frac{\psi_{base-run_i} - \psi_{perturbed_i}}{\theta_{base-run_i,k} - \theta_{base-run_i,k} \cdot 1.01}, \tag{1}$$

where $\psi$ denotes the model output that is evaluated, $\theta$ refers to the parameter value, $k$ the number of parameters that is evaluated in the sensitivity analysis (in our case 15, 18, and 17 for HBV, SAC, and VIC, respectively), and $i$ the number of base runs (in our case 100). In total, this leads to 1500+100, 1800+100, and 1700+100 runs per basin for the three hydrological models, totalling to 9,619,500 model runs considering the forcing of three different GCMs.

We used the average sensitivity from the 100 samples per parameter per basin as a measure of parameter sensitivity. Each parameter that is displayed in Figure 2 was accounted for in the sensitivity analysis. The applied parameter boundaries for sampling are provided in Appendix Tables A1, A2 and A3 (see also Melsen and Guse, 2019).

Besides the selection of a sensitivity analysis method (which will influence the final results, Razavi and Gupta, 2015; Pianosi
et al., 2016), we also had to identify a target variable - the variable that is compared between the base-run and the perturbed run. Whereas performance metrics are quite popular as target variable (Van Werkhoven et al., 2008; Herman et al., 2013a), they are not well suited for global sensitivity analysis (Razavi and Gupta, 2015; Guse et al., 2016a), and besides, it is not possible to obtain model performance for the future. Therefore, this study focuses on mean simulated discharge and the day of the year that half of the discharge has passed (hereafter referred to as 'discharge timing') as target variables. Many other streamflow
signatures could have been of interest to evaluate, for instance related to high and low flows, but given the goal of this study, an exploration on the effect of climate change on parameter sensitivity, a volume and a timing signature based on discharge seem the most neutral choice.

The sensitivity analysis was conducted for both the historical and future period, for all 605 basins forced with three different
plausible climate change rates (based on the three GCMs). The first analysis of the calculations was a simple exploration of which parameters increase and which parameters decrease in sensitivity in the future over all 605 basins to achieve a first insights into potential changes in parameter sensitivity in future and to see which parameters are mainly affected.

## 2.4 Climate indicators to relate changes in sensitivity

The 605 climate instances from the 605 basins are not a representative sample since certain climates might be over- or un-
170 derrepresented. Therefore, the difference in sensitivity was also related to climate indicators. This also allows to combine the results obtained with the three different GCMs: if there is a relation between a climate indicator and parameter sensitivity, this should be visible regardless of which GCM was used. Basically, the three GCMs were used to sample the plausible climate change space.

Given their relevance for discharge, we employed the Knoben climate indicators (Knoben et al., 2018) to classify the changes in parameter sensitivity. The Knoben indicators consist of three indicators: aridity index, seasonality, and fraction precipitation falling as snow.

To determine the aridity index, first Thorntwaite's Moisture Index ($MI$, Knoben et al., 2018; Willmott and Fedema, 1992) is obtained based on mean monthly observations of precipitation, $P(t)$, and evapotranspiration $E_p(t)$. Subsequently, average aridity $I_m$ can be obtained.

$$MI(t) = \begin{cases} 1 - \frac{E_p(t)}{P(t)}, & \text{if } P(t) > E_p(t) \\ 0, & \text{if } P(t) = E_p(t) \\ \frac{P(t)}{E_p(t)} - 1, & \text{if } P(t) < E_p(t) \end{cases} \tag{2}$$

$$I_m = \frac{1}{12} \sum_{t=1}^{t=12} MI(t) \tag{3}$$

Aridity index $I_m$ varies between -1, representing highly arid conditions, and 1, representing humid conditions. The seasonality in the aridity index, $I_{m,r}$, is determined based on the maximum difference in $MI$ over the year:

$$I_{m,r} = \max(MI(1, 2, ...12)) - \min(MI(1, 2, ...12)) \tag{4}$$

The seasonality varies between 0 and 2, with 0 indicating no intra-annual variation, and 2 indicating that climate varies from fully arid to fully humid over the year. The last Knoben index is the fraction precipitation falling as snow, $f_s$.

$$f_s = \frac{\sum P(T(t) \leq T_0)}{\sum_{t=1}^{t=12} P(t)} \tag{5}$$

In this equation, $T(t)$ is the mean monthly temperature, and $T_0$ the threshold temperature below which precipitation is assumed to occur as snow. The threshold temperature was set to $0°$C, in line with Knoben et al. (2018). $f_s$ can have a value between 0, no snow, and 1, all precipitation falling as snow. All three indicators were evaluated based on the climate in the historical period, and based on the change in climate between the future and the historical period (future - historical), this latter one being indicated as $\Delta$indicator.

## 2.5 Evaluation of impact of sensitivity changes on calibration strategy

To evaluate the impact of change in parameter sensitivity on calibration strategy, we determined the top-5 most sensitive parameters for each basin, both for the historical and future period. We analyzed which parameters left and entered the top-5 in the future compared to the historical period, as a consequence of a change in sensitivity. This was again related to the climate indicators of Sect. 2.4, to investigate if in certain climates or climate change rates more changes can be expected in the calibration parameters.

## 2.6 Diagnostic model evaluation based on changes in sensitivity

To diagnose how the results from the different models have come about, we relate the direct model output (several states and fluxes) to changes in sensitivity. Furthermore, we introduce the concept of 'parameter sensitivity transmission': We evaluate

whether any negative correlations exist between parameters with increasing and decreasing sensitivity. Strong negative correlations can be an indication that sensitivity is transmitted from one parameter to the other, so we define transmission as a clear negative correlation in change in sensitivity between two parameters. However, since we evaluate correlation, transmission does not refer to absolute sensitivity values.

The goal of this analysis is to investigate to what extent sensitivity is transmitted directly from the decreasing parameter to the increasing parameter. When there is no direct relation, it can indicate that sensitivity changes at several places within the model structure. The transmission of sensitivity can give insights into which processes become more relevant in the future, at the expense of processes that become less relevant - a systemic change as a result of climate change. A comparison among the different model structures will indicate their (dis)agreement on the change in relevant processes.

## 3 Results

To place the results into context, we first briefly discuss the change in climate between the historical and future period, and the changes in several simulated water balance terms for both periods for the three employed hydrological models. Subsequently, we discuss the change in sensitivity between the historical and future period, the relation between changes in sensitivity and climate, the impact of sensitivity changes on calibration strategy, and finally the model diagnostic evaluation.

### 3.1 Changes in climate and simulated water balance terms between historical and future period

Fig. 3 provides an overview on the change in climate (expressed in temperature and precipitation) between the historical and future period for the three employed GCMs (RCP8.5). In all the investigated basins the mean temperature will increase in the future with at least 2.6°C. IPSL clearly simulates a warmer climate than the other two GCMs. IPSL also shows a decrease in precipitation for half of the basins, whereas most basins will receive more precipitation in the future in the MPI and CCSM projections.

Fig. 3 also depicts several simulated water balance terms for the three employed hydrological models (subdivided for the three different GCMs). Note that the hydrological models were not calibrated and that for each of the 605 basins, the mean across the parameter ensemble was used for this figure. It is therefore not an indication of what exactly might happen in the future in the investigated basins, but an indication of the flexibility of the models in responding to changes in climate. There is consistency among the models that in most basins, evapotranspiration will increase in the future. Here, the differences are larger among the GCMs than among the hydrological models. There is also agreement among the hydrological models that snow water equivalent will decrease in the vast majority of the basins, although the magnitude of change differs among the hydrological models - more than across the GCMs. The signal in discharge has a less clear direction, which is also consistent among the models, although VIC seems to hinge on a general increase in discharge while HBV and SAC have slightly more

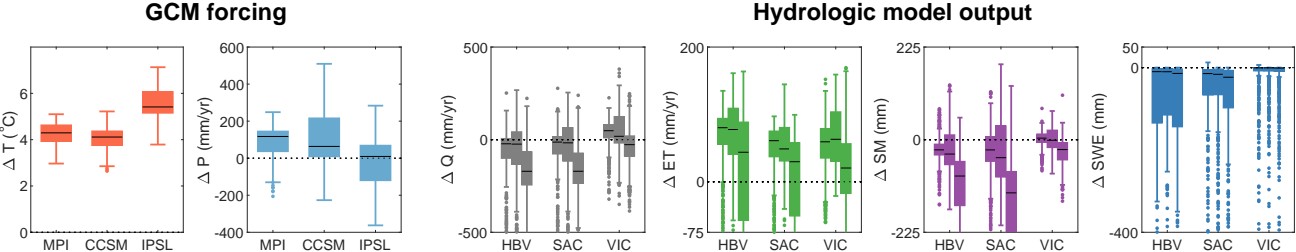

**Figure 3.** Changes between historical (1985-2008) and future (2070-2093) period. The left two panels depict the change in temperature (ΔT) and precipitation (ΔP) across the 605 basins, as obtained with MPI, CCSM, and IPSL (RCP8.5). The right four panels depict the change in mean discharge (ΔQ), mean evapotranspiration (ΔET), mean soil moisture (ΔSM) and mean snow water equivalent (ΔSWE) between both periods for the 605 basins, as simulated by the three different hydrological models and forced with the three different climate models. Each model was run for a full parameter sample per basin, the average change across the parameter sample per basin was used to create the boxplots.

basins where discharge would decrease. Both HBV and SAC simulate a decrease in soil moisture in most basins, especially in the IPSL forced runs, whereas the decrease in soil moisture for the VIC simulations is much lower. The hydrological models seem to broadly agree on the general direction of change in several of the simulated water balance terms, but differences among hydrological models can already be observed and might be more pronounced for individual basins. For soil moisture and evapotranspiration, the GCM forcing seems more relevant to explain the differences, while for snow water equivalent, differences are mainly found between the hydrological models.

### 3.2 Changes in sensitivity between historical and future period

Fig. 4 and Fig. B1 show the distribution of change in sensitivity between the historical (1985-2008) and future period (2070-2093) over all 605 basins for the three hydrological models for three GCMs, for mean discharge and discharge timing as target variable, respectively.

Consistent over all three hydrological models when evaluating the mean discharge as target variable, is a decrease in the sensitivity of snow parameters in the future. The parameters that show increasing sensitivity cannot consistently be associated to one specific process. Whereas a strong decrease in sensitivity requires a high sensitivity in the historical period, this is not required for a strong increase in sensitivity. It can be observed, however, that model parameters that display an increase in sensitivity were also already sensitive in the historical period.

In the HBV model especially the snow correction factor (SCF) displays a large decrease. This is also the parameter with the highest sensitivity in the historical period, therefore having the highest potential to decrease. The other three snow parameters in HBV displayed lower sensitivity in the historical period, and also show a less consistent decrease in the future. Also in

the SAC and VIC models, the snow parameter that displayed the highest sensitivity in the historical period (SCF in SAC and Snowrough in VIC, respectively) show the strongest decrease.

Among the three hydrological models, different parameters related to different processes display an increase in sensitivity in the future. In HBV, evapotranspiration and soil parameters increase in sensitivity in the future with the largest increase in the evapotranspiration parameter PT, while there is hardly any observable change in sensitivity in percolation and deep layer parameters. In the SAC and VIC model, there are parameters associated to all processes except snow, that tend to mainly increase in sensitivity in the future. Like for HBV, also in SAC the evapotranspiration parameter PT has the highest increase. In the VIC
model, the depth of the second soil layer (Depth2) shows the largest positive change in sensitivity. Consistent with the results in the previous section on water balance terms, it can be seen that changes in sensitivity are quite consistent among the three different GCMs when it considers snow related parameters, whereas more differences can be observed in parameters related to soil moisture and evapotranspiration processes.

Fig. B1 shows that for the discharge timing, many more processes in the hydrological models display changes in sensitivity. The decrease in sensitivity of snow parameters is less pronounced although still present, while many other parameters show small increases in sensitivity. This is consistent among the three hydrological models, and consistent among the three GCM forcings.

### 3.3    Relationship between climatic variables and sensitivity changes

Since the 605 basins employed in the previous section are not a representative, balanced sample over climates and climate changes, the results are split out over the three Knoben climate indicators. Fig. 5 depicts how parameter sensitivity changes between historical and future period for six relevant model parameters for the mean discharge, Fig. B2 for discharge timing. The results obtained from the three different GCMs have been combined, because they can be perceived as different samples of plausible climate change space.

From Fig. 5, it can be seen that the patterns relating parameter sensitivity to climate and climate change indicators are weak, except for evapotranspiration parameters. The projected change in aridity index seems to have explanatory value for the change in sensitivity of evapotranspiration parameters PT (both in HBV and SAC) and Rmin (VIC). The change in sensitivity of snow parameters cannot be explained with the Knoben climate indicators, but can be related to current mean temperature and pre-
cipitation, and projected changes in mean temperature (Fig. A1). The patterns, however, vary per model.

In most cases, the patterns that can be identified relate to the projected change in climate. For instance, soil moisture/shallow layer parameter Depth2 (VIC) and percolation parameter Expt2 (VIC) demonstrate a more pronounced increase in regions with decreasing aridity index. Sometimes also the historical climate, combined with the projected change, can show organization.
For example, the sensitivity of the evapotranspiration parameter PT in both SAC and HBV is particularly increasing in regions

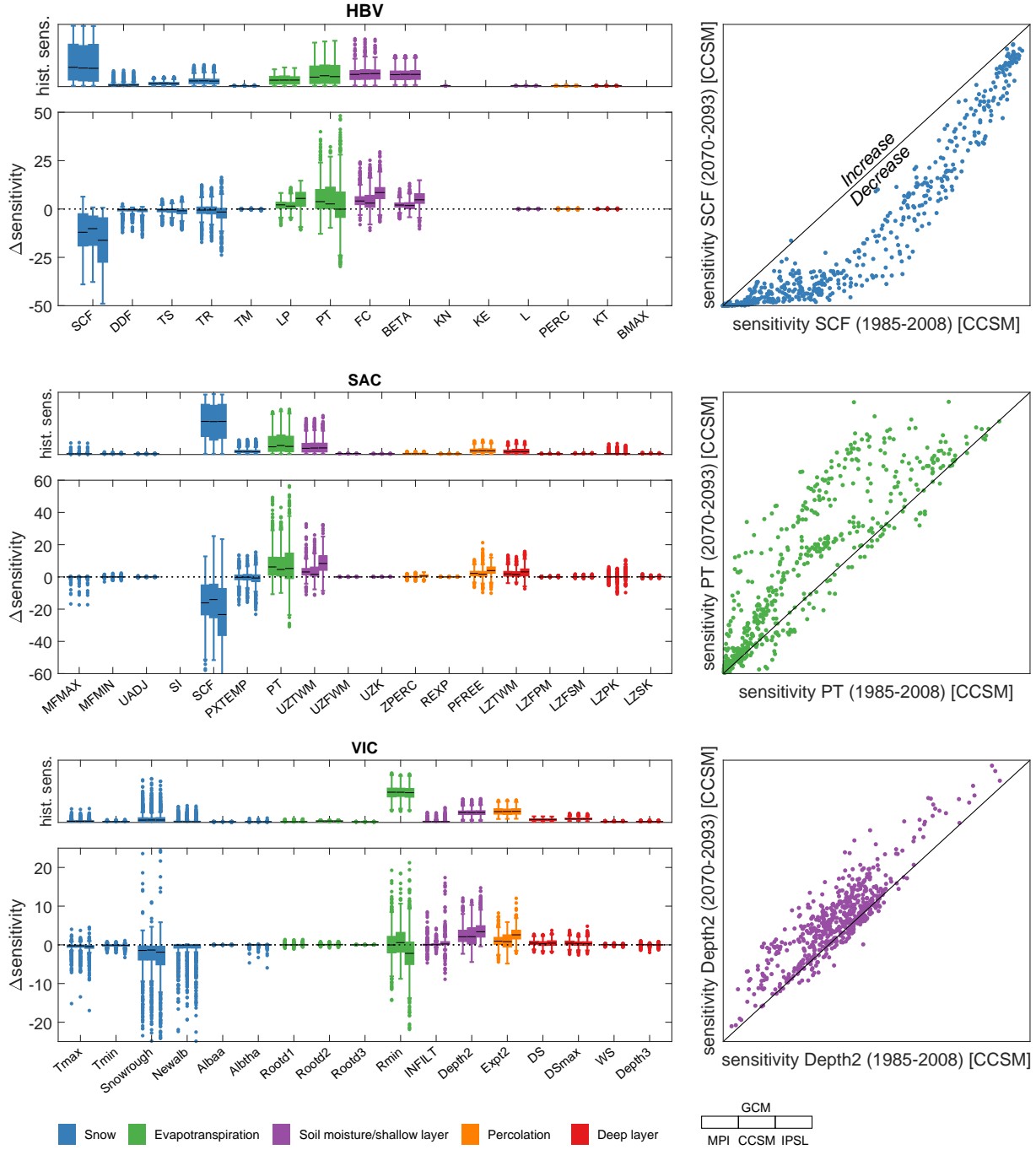

**Figure 4.** The distribution of change in parameter sensitivity (Δsensitivity) over 605 basins for the period 2070-2093 compared to 1985-2008, displayed per parameter per hydrological model, for three different GCM forcings. Above each Δsensitivity panel, historical sensitivity is displayed. The panels on the right show the data for a selected case per model.

with high historical aridity index, and the direction of change relates to the change in aridity index.

Given that no clear patterns were revealed based on the Knoben indicators, we also explored patterns related directly to climate: the mean temperature and mean precipitation and their projected changes. These results for mean discharge as target variable can be found in Figure A1 in the Appendix. The snow parameters mainly decrease in sensitivity in basins with a historically mean temperature between 0 and 15°C, dependent on the model. In these basins, the fraction of snow will decrease in a warmer climate, whereas in basins with a lower mean temperature, snow will remain a relevant process in the future (Fontrodona Bach et al., 2018). Also the evapotranspiration parameters (PT for HBV and SAC, and Rmin for VIC) demonstrate a clear relation with temperature and precipitation, in the same fashion as was found for the aridity index.

With discharge timing as target variable, we found overall similar sensitivity patterns (Fig. B2 and B3): change in aridity shows most organisation when related to change in sensitivity, and change in snow parameters can be related to historical temperature. However, the absolute values of the changes in sensitivity for discharge timing are lower compared to the sensitivity changes in mean discharge.

## 3.4 Impact of sensitivity changes on calibration strategy

In this section we explore to what extent the changes in parameter sensitivity that were observed in the previous sections propagate into the calibration procedure. To this end, we evaluate the top-5 most sensitive parameters, and how this top-5 changes between the historical and future period.

Fig. 6 depicts which parameters leave the top-5 in the future, and which parameters enter the top-5 in the future for the mean discharge as target variable. Mainly snow parameters drop out of the top-5 in all three models, while parameters related to many other processes enter the top-5. Although changes in top-5 parameters are observed, the overall top-5 of the parameters is maintained in 45% to 55% of the catchments, dependent on the hydrological model. In 41 to 49% of the catchments, one parameter changes in the top-5, in 3 to 6% of the catchments two parameters change in the top-5. The maximum number of changes in the parameter top-5 per catchment is three, which occurs only in max 0.2% of the investigated basins.

For HBV, especially snow parameters SCF and TR exit the top-5 in the future runs. The largest increase in top-5 notations for HBV is found for evapotranspiration parameter LP. Remarkably, snow parameter TR (threshold temperature where precipitation falls as rain) also appears as a parameter that enters the top-5: this parameter leaves the top-5 in many basins, but also enters the top-5 in many other basins. In SAC, snow parameter PXTEMP loses the most top-5 notations. Lower zone parameter LZTWM shows the strongest increase in top-5 notations. In VIC, mainly the snow parameter Snowrough decreases in top-5 notations. Deep layer parameters gain most notations, especially DS.

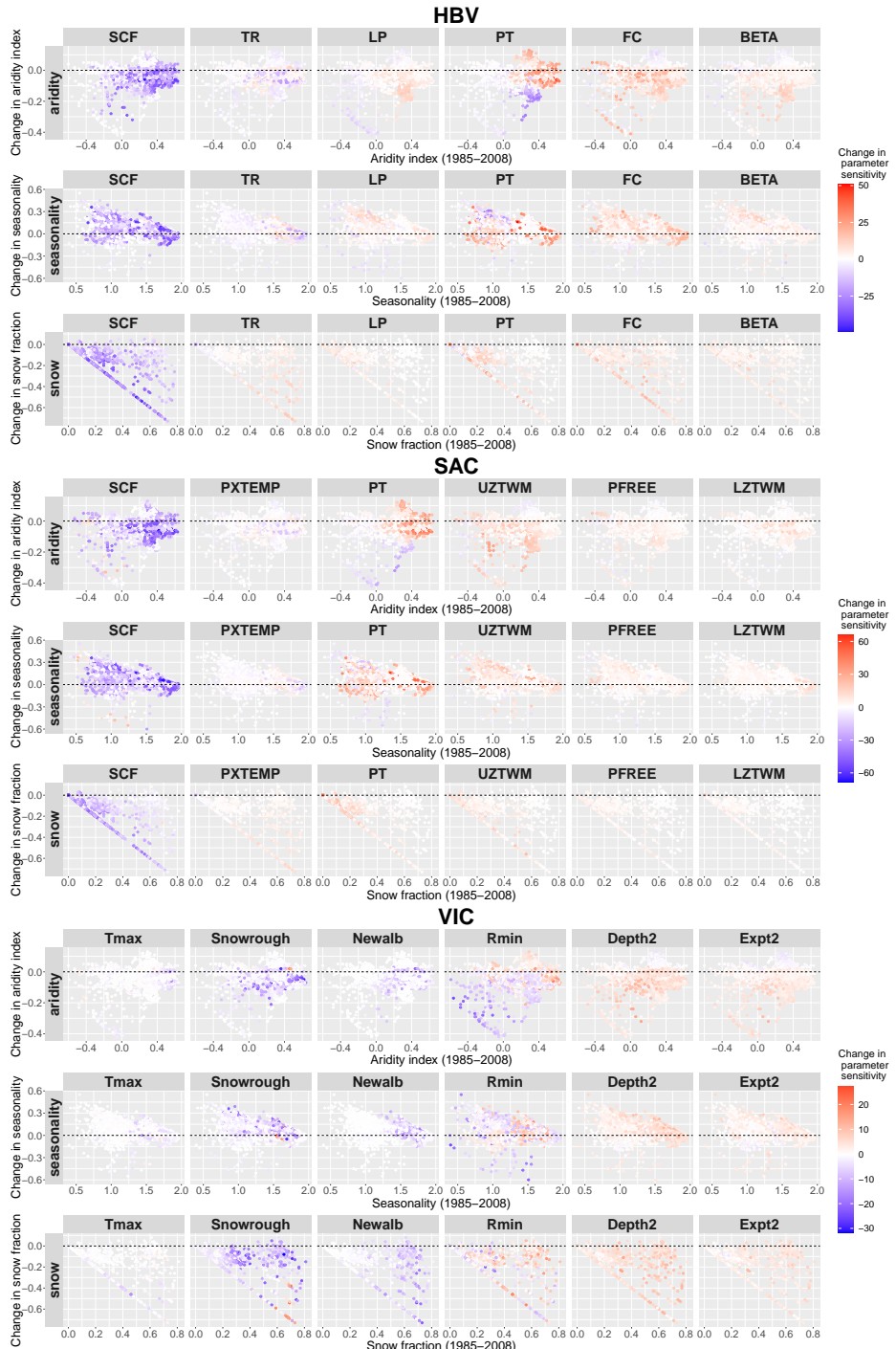

**Figure 5.** Change in parameter sensitivity versus historical climate indicators and change in climate indicators for 605 basins. The climate indicators are determined for all three GCMs. Displayed are aridity index (-1 highly arid, +1 highly humid), seasonality, and fraction of precipitation falling as snow, as defined by Knoben et al. (2018). Parameter sensitivity for the historical period is expressed in dot size, change in parameter sensitivity in colour: red indicates an increase in sensitivity, blue a decrease.

The results of the pie charts in Fig. 6 cannot directly be generalized because the 605 explored basins are not a well-balanced sample in terms of climate and climate change. Therefore, the change in top-5 parameters is also again displayed against the Knoben indicators (right panels in Fig. 6). It can be observed that one change in parameter top-5 can occur across all climates and climate changes. Two changes in parameter top-5 already show some clustering, at least for HBV and VIC this seems to be related to a high historical aridity index (wet conditions). For HBV, also a cluster of two and three parameter changes can be found in the catchments that display high seasonality.

The same results are displayed in Fig. B4 but then for the discharge timing as variable of interest. Also for the discharge timing, across all three models mainly the snow parameters leave the top-5, although also parameters related to other processes leave top-5 positions - more than for mean discharge where snow parameters really dominate in losing their top-5 position. The type of parameters entering top-5 positions varies a lot among the three models; evapotranspiration parameters for HBV, percolation parameters for SAC, and soil moisture/shallow layer parameters for VIC. Also the change in top-5 positions in general varies highly between the models; for VIC, 61% of the catchments does not experience any change in parameter top-5, while in SAC, only 12% of the catchments maintains the same top-5 in the future. Two changes in the parameter top-5 are more frequent than for mean discharge (varying between 7 and 34%). There are even some exceptional cases (0.06% in HBV) where the complete parameter top-5 has changed.

In conclusion, changes in parameter sensitivity as a consequence of climate change can propagate into the calibration strategy. For the mean discharge, one change in the parameter top-5 is common and can occur across all climates and climate changes, whereas about 4% of the explored catchments experiences at least two but max three changes in parameter top-5. As such, the impact seems limited. This is, however, very much dependent on the signature that is evaluated, and the hydrological model employed. For the discharge timing (defined as the day of the year that half of the discharge has passed) as target variable, changes in the top-5 are much more common for some models, and up to 5 changes in the top-5 were found for exceptional cases.

### 3.5 Diagnostic model evaluation based on changes in sensitivity

The evaluated changes in parameter sensitivity in response to climate change can be perceived as a way to evaluate models diagnostically, especially since we can compare the results for three different hydrological models. The parameter sensitivity in the historical period (the top panels in Fig. 4) already shows that the models activate different processes to simulate historical discharge. Our analysis of change in sensitivity demonstrates that the models also respond differently to changes in forcing.

There are a few points where all three hydrological models agree: all models simulate a decrease in snow in the future across all basins, and an increase in ET across most basins (Fig. 3). This is also visible in the change in sensitivity for the mean discharge of the parameters related to these processes. In all models, the snow parameters tend to decrease most in sensitivity (dependent on their historical sensitivity), and a tendency to increase in sensitivity for ET parameters was found (although clearly weakest

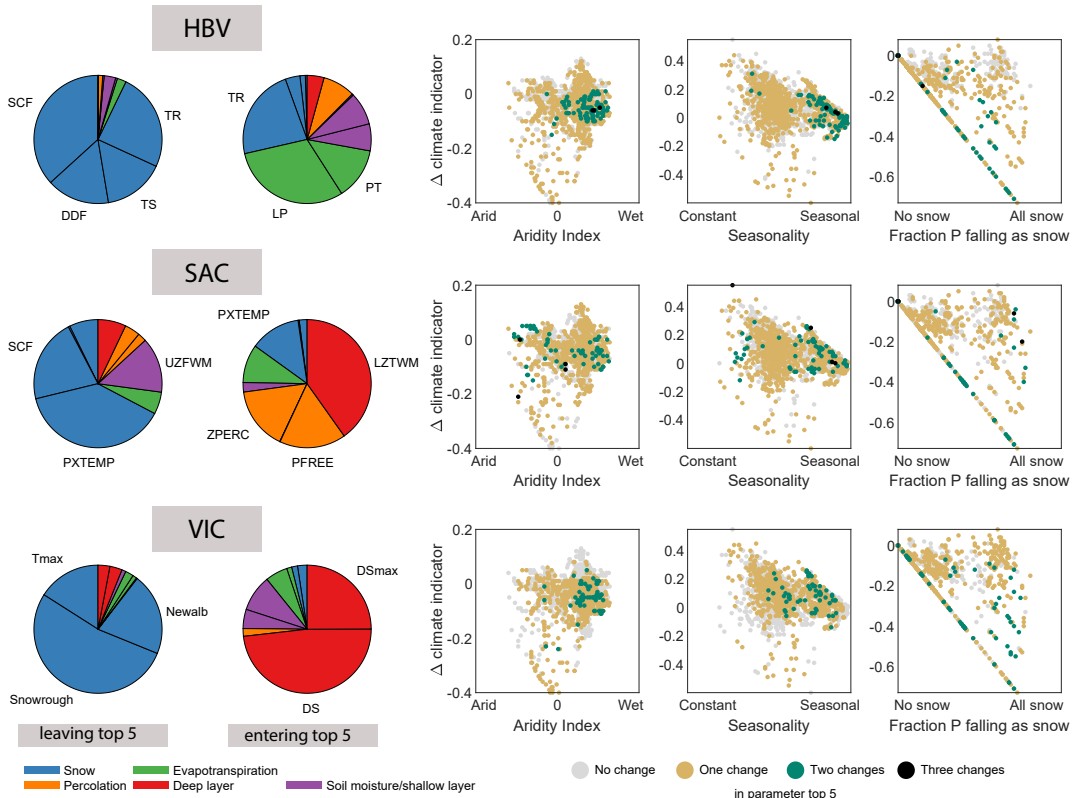

**Figure 6.** Impact of change in parameter sensitivity on top-5 position for the mean discharge, where top-5 refers to the five most sensitive parameters per basin - generally the parameters that are calibrated. The pie charts show which parameters leave the top-5 (left) and which parameters enter the top-5 (right). The right panels relate the number of changes in the parameter top-5 to climate and climate change indicators.

for VIC, Fig. 4, and dependent on GCM). These results are robust across different formulations for snow and ET processes: SAC and HBV share the same ET formulation and employ a comparable snow formulation, but VIC employs a completely

different formulation for both ET and snow. Yet, all three models agree on these signals.

However, many other changes in sensitivity for mean discharge can be observed where the models disagree, for instance the role of percolation and soil moisture/the shallow layer. To further explore how the models respond to climate change in terms of parameter sensitivity, the transmission of sensitivity is explored by means of the negative correlation between change in

sensitivity among two parameters. An example is the left panel of Fig. 7, depicting a negative correlation between the change in sensitivity of snow parameter SCF and the change in sensitivity of evapotranspiration parameter LP for HBV (combining all three GCM forcings), which can indicate a transmission of sensitivity from SCF to LP. The chord diagrams in Fig. 7 show the

correlations between the parameters with decreasing and increasing sensitivities. All three models display a decrease in sensitivity of the snow parameters, but this sensitivity is transmitted to different process parameters in the three models. In HBV, mainly to evapotranspiration and shallow layer parameters, in SAC evapotranspiration, percolation, and deep layer parameters, and in VIC to shallow layer and deep layer parameters. Weak transmissions can indicate that parameter sensitivity changes at several places in the model structure.

Fig. B5 displays the chord diagram for the discharge timing as target variable. The results are very comparable to the results for the mean discharge: mainly the snow parameters decrease in sensitivity, and this sensitivity is transmitted to parameters representing different processes across different hydrological models: Evapotranspiration and shallow layer parameters in HBV, percolation and deep layer parameters in SAC and mainly deep layer parameters in VIC. Interesting is the role of shallow layer parameter INFILT in VIC. This parameter transmits sensitivity to three different deep layer parameters. But in Fig. B4, it is also visible that INFILT is one of the parameters that frequently enters the top-5 in the future and therefore this parameter could also be displayed at the top of the chord diagram. There is, however, no parameter in VIC that displays a clear negative correlation in sensitivity change with INFILT: the strongest correlation that was found was -0.16, between snow parameter Newalb and INFILT. INFILT thus mainly transmits sensitivity to deep layer parameters, but gains sensitivity from several different sources.

Whereas the models agree on the decline in snow water equivalent and decreased sensitivity of snow parameters despite employing different snow formulations, the models disagree on changes related to many other processes. Since the three models differ in many aspects in their model structure, the difference in response to changing forcing cannot directly be related to specifics of the model structure. The results, however, do show that the internal functioning of the models differ when used for long term simulations, and this might impact the results and subsequently the conclusions of the model study.

## 4 Discussion

### 4.1 Changes in states, fluxes, and sensitivity between historical and future period

A first evaluation of the different states and fluxes that are simulated by the hydrological models for the future period demonstrates that the three models agree that in general, snow water equivalent will decrease under RCP 8.5 using three different GCMs. This same signal is propagated into the sensitivity of the parameters related to this process: in all three models, the sensitivity of snow parameters tends to decrease for the mean discharge. All models also agree on the tendency that evapotranspiration will increase in the future, although this varies across GCM forcing. Also this signal is reflected in the sensitivity of evapotranspiration parameters for mean discharge: their sensitivity tends to increase (although the models disagree on the magnitude of change). These results imply that the impact of snow on mean discharge will decrease, while the impact of evapotranspiration on mean discharge will increase. The results for discharge timing are much more variable across the models.

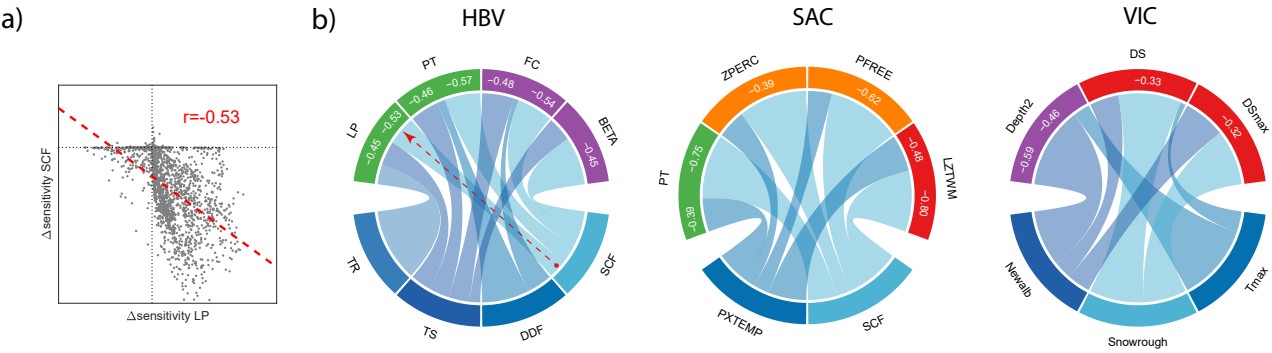

**Figure 7.** Indication of parameter sensitivity transmission. Panel a) an example for HBV: the change in sensitivity of parameter SCF shows a strong negative correlation with the change in sensitivity of parameter LP, which can indicate that SCF transmits sensitivity to LP. Since we focus on transmission, we only evaluate negative correlations. Panel b) The chord (circle) diagrams display transmission of sensitivity, indicated with a band from the parameter that decreases in sensitivity to the parameter that increases in sensitivity. The width of the band indicates the strength of the negative correlation. The example from panel a is indicated with a red arrow in the chord diagram of HBV. The white number indicates the strength of the correlation; -0.53 between SCF and LP. In all three chord diagrams, the lower part shows the parameters that decrease in sensitivity, and the upper part the parameters that increase in sensitivity, with the white number indicating the strength of the correlation (for clarity, lower correlations are not displayed). Colors are according to the process they represent (with different shades of blue used for snow parameters for clarity). The chord diagrams are focused around the most relevant parameters based on Fig. 6.

For other states and fluxes simulated by the models, such as soil moisture and percolation, the models agree less on the change in sensitivity when using mean discharge as target variable (Fig. 4). This can first and foremost be attributed to a difference in model structure, but the impact of model structure can be further emphasized by the target variable that we used for our sensitivity analysis. We evaluated the sensitivity of parameters to simulate mean discharge. For instance for HBV, percolation parameters historically already did not display a strong sensitivity for mean discharge, and the sensitivity of percolation parameters does not change in the future. This does not automatically imply that HBV does not simulate a change in percolation as a consequence of climate change, but mainly that mean discharge and percolation are decoupled (at least in comparison to other processes) in the HBV model structure. Another example is that all models simulate a substantial increase in evapotranspiration, but for VIC this does not lead to a substantial increase in sensitivity of parameters related to ET. Other signatures as target variable therefore will lead to different results: Fig. B1 demonstrates that when using the timing of the discharge as target variable, both historic sensitivity and change in sensitivity is substantially different from the results for the mean discharge. The results of this study should thus be seen conditional on the explored target variables: for studies focusing on long-term projections of flood and drought, flood and drought specific variables should be explored to investigate changes in sensitivity.

## 4.2 Climate indicators to relate changes in sensitivity

We evaluated change in sensitivity against three climate indicators; aridity index, seasonality, and fraction precipitation falling as snow. We were not able to identify a clear, robust relation between climate indicator, change in climate indicator, and change in parameter sensitivity. In our approach, we investigated if any temporal relations exist. Another way to evaluate change in sensitivity would be to evaluate spatial relations. van Werkhoven et al. (2008) for instance, demonstrate that spatial gradients in model sensitivity exist that relate to climate. If we can establish a temporal relation in the same way van Werkhoven et al. (2008) could demonstrate spatial relations, space-for-time trading would be an option to determine which parameters become sensitive in the future. The lack of a clear relation between climate, climate change, and parameter sensitivity however, also demonstrates that we should critically evaluate the adequacy of the model structures for long-term projections.

## 4.3 Impact of sensitivity changes on ranking of sensitive parameters

We investigated how parameter sensitivity changes as a consequence of climate change. We also explored the use of sensitivity analysis to provide the most relevant parameters (factor prioritisation) for an effective model calibration (Saltelli et al., 2006; Reusser et al., 2011). Within this context, we have shown how changes in parameter sensitivity propagate into the selection of relevant parameters for model calibration. We assumed a general calibration strategy where the modeller selects the five most sensitive parameters for calibration. Certainly many other calibration strategies exist. For example, one could select all the parameters that exceed a certain sensitivity-threshold as suggested by van Werkhoven et al. (2009) or when compared to a dummy parameter as suggested by Zadeh et al. (2017), resulting in a higher or lower number of parameters for the calibration, or simply include all the parameters in the model if the model is highly parsimonious (Melsen et al., 2014). Our results are, however, still relevant in the context of other calibration strategies, as the changes in sensitivity will still influence the calibration results. That is, it is difficult to calibrate a parameter if the model is hardly or not sensitive to changes in its values in current-day climate.

The implication of our result is that, the more the parameter sensitivity changes, the more parameter identifiability decreases for long-term projections. Accordingly, we can expect that in particular the parameters that will enter the top-5 in the future are probably not well identified in the historical period. Therefore, we provide suggestions to account for changing sensitivities in the calibration strategy of hydrological models for long-term projections. A first strategy, related to methods that have been suggested for changing parameters over time (Merz et al., 2011; Vaze et al., 2010), is to conduct sensitivity analysis over different parts of the observation-period, and calibrate the model on the period that best resembles the parameter sensitivity of the future scenario. A risk, however, is that the calibration period becomes too short to determine stable parameter values (Yapo et al., 1996). A second strategy is to sample the parameters that will become sensitive in the future. Provided that, in the current climate, the model is not sensitive to changes in parameter $\theta_i$, the value of $\theta_i$ cannot be inverted through calibration. However, in the future the value of $\theta_i$ does become relevant. In order to correctly capture spread in long-term projections that results from uncertainty in this parameter-value, the value of $\theta_i$ should be sampled. Hereby, we have to emphasise that in this

context, parameter uncertainty is specifically related to expected changes in the relevance of the associated processes in future. A third option could be to increase the effort in finding data to be able to calibrate a parameter directly to the associated process. Sensitivity analyses on different processes demonstrate that the sensitivity signal increases using the associated process as target variable (Guse et al., 2016a). In this way, it can be expected that parameters are better identifiable and more robust for future simulations.

## 4.4 Diagnostic interpretation

In the previous sub-section we provide suggestions to further validate the calibration procedure of models employed for long-term projections. It seems a valid question, however, whether our models are fit for this purpose at all. The results of the sensitivity analysis indicate a change in relevant processes in the future which is captured differently among the three investigated models. This emphasizes the need to improve model structure for long term projections, as suggested by Fowler et al. (2018), Grigg and Hughes (2018) and Westra et al. (2014).

Assuming that a sensitivity analysis conducted over 23 years of daily data is robust and thus that the observed changes in sensitivity can be attributed to a changing climate rather than to noise, our results demonstrate that parameter sensitivity is nonstationary (Koutsoyiannis and Montanari, 2015). Nonstationarity of parameter sensitivity fits in the growing body of literature identifying nonstationarity when simulating the hydrological system on the long term (e.g. Milly et al., 2008; Thirel et al., 2015; Fowler et al., 2016, 2018). Nonstationarity is not only disclosed through a change in sensitivity, but also through a change in parameter values over time (Vaze et al., 2010; Merz et al., 2011). The identification of nonstationarity in parameter values is the result of the simplified model representations, not capturing dynamics and/or processes that are relevant in the real world. Fowler et al. (2018) provides a framework to evaluate model improvement under nonstationary conditions; Grigg and Hughes (2018); Westra et al. (2014) and Duethmann et al. (2020) adapted model structure to account for nonstationarity, leading to improved model results. This study reinforces this direction of research; even though the decrease in sensitivity among all three models can consistently be found for the snow parameters, the increase in sensitivity can be attributed to different processes in the three models, which might indicate that a relevant process is missing in any of the models, stressing the need to carefully assess whether these models are appropriate for long-term projections. The differences in which processes and associated parameters becomes more relevant among the models shows that there is no consensus how the hydrological system will change in future.

A decrease of sensitivity of snow parameters and an increase in the sensitivity of evapotranspiration parameters in a warming climate (considering enough moisture being available) could be expected based on expert judgement, and at least the three models agree on those signals despite employing different formulations to compute these processes. However, the models disagree on the other processes that will become more or less relevant in the future, while changes in these processes are not straight forward to estimate based on expert judgement. It is, for instance, not easy to judge whether the relatively higher

amount of rain in the future (due to a decrease in snow) goes on average more to higher evaporation or to higher infiltration. As such, we have to acknowledge that the models differ in the processes they use to simulate future changes, and that we cannot easily differentiate the right from the wrong models. This calls for a more process-based evaluation of historical changes, to evaluate their plausibility for future changes to guide model selection.

## 5   Conclusions

In this study we investigated if hydrological model parameter sensitivity changes within a plausible climate change rate. This is relevant for parameter prioritization in the calibration procedure for long-term projections, and can be insightful for model diagnostic evaluation to investigate how the models simulate systemic changes as a consequence of climate change.

The sensitivity of the parameters in three investigated hydrological models changes within a plausible changing climate. The three models agree that especially the snow parameters decline in sensitivity, while evapotranspiration parameters show a tendency to increase (dependent on the employed GCM). Which other parameters increase in sensitivity is, however, less consistent among the models; sometimes mainly ET and soil moisture/shallow layer parameters, sometimes mainly percolation and/or deep layer parameters. These differences occur due to differences in the three hydrological model structures. We did not identify a clear pattern in which kind of climates and expected climate changes most changes in parameter sensitivity take place.

The change in parameter sensitivity propagates into the calibration strategy. Typically, a global sensitivity analysis is conducted to determine the most sensitive parameters, and based on that, the top-5 most sensitive parameters are selected for calibration. Dependent on the model, 45% to 55% of the 605 investigated catchments has at least one parameter changing in the top-5 in the future when mean discharge is evaluated, and between 40% and 88% of the catchments when discharge timing is evaluated. For the mean discharge, the highest number of changes in the parameter top-5 was three, while for discharge timing, in some exceptional cases the complete top-5 consisted of different parameters in the future. Since these results were obtained for the highest emission scenario (RCP8.5), fewer changes might be expected for lower emission scenarios.

Some parameters become sensitive in the future, but are currently not sensitive. Therefore, their value cannot be obtained through calibration based on current data. One way to account for changes in sensitivity is to identify a historical period that mimics the future projected sensitivity. Another approach is to sample the parameter that becomes sensitive in the future, to account for predictive uncertainty as a consequence of the uncertainty in this parameter value. A third approach is to invert the value of this parameter based on observations specifically related to the process that the parameter is related to.

Besides implications for the calibration strategy when using models for long-term projections, our results also have implications for model selection for this purpose. The results demonstrate that the three employed models consider different processes as becoming more or less relevant in the future; they simulate different systemic changes. Whereas the models agree on

systemic changes that can be excepted based on expert judgement (decreased relevance of snow and increased relevance of evapotranspiration in a warming climate), the models disagree on other processes that are more difficult to judge. These results not only stress the need, but also the challenge in carefully assessing model structure adequacy when applying models for long-term projections.

## Acknowledgments

BG thanks for financial support from the German Research Foundation ("Deutsche Forschungsgemeinschaft", DFG) via the FOR 2416 "Space-Time Dynamics of Extreme Floods (SPATE)" research group. The authors would like to thank three anonymous reviewers for their constructive feedback, and Thorsten Wagener for his suggestions.

*Data availability.* Results from the sensitivity analysis can be downloaded from: http://www.hydroshare.org/resource/36e195451798497193628c235b537052

*Author contributions.* LM and BG designed the study together. LM conducted the calculations and wrote the first draft of the paper. LM and
525 BG processed the data together and finalized the manuscript together.

*Competing interests.* There are no competing interests

## Appendix: A. Parameter ranges

**Table A1.** Selected parameters, their classification, and their boundaries for the HBV model. The parameters and their boundaries are based on Parajka et al. (2007); Uhlenbrook et al. (1999); Abebe et al. (2010). The Priestley-Taylor parameter is based on Lhomme (1997).

|    | Name | Unit | Lower boundary | Upper boundary | Description |
|----|------|------|----------------|----------------|-------------|
| 1  | Tm   | °C   | -3.0    | 3.0    | Temperature where snow melt starts |
| 2  | Ts   | °C   | Tr-0.01 | Tr-3   | Temp. below which precipitation is snow |
| 3  | Tr   | °C   | 0.0     | 3.0    | Temp. above which precipitation is rain |
| 4  | DDF  | mm °C$^{-1}$ d$^{-1}$ | 0.04 | 12 | Degree day factor |
| 5  | SCF  | -    | 0.1     | 5.0    | Snow correction factor |
| 6  | LP   | -    | 0.0     | 1.0    | Evaporation reduction threshold |
| 7  | PT   | -    | 1.0     | 1.74   | Priestley-Taylor coefficient |
| 8  | FC   | mm   | 0.0     | 2000   | Max soil moisture storage |
| 9  | BETA | -    | 0.0     | 20     | Non-linear shape coefficient |
| 10 | K0   | day  | 0.0     | 2.0    | Storage coefficient of very fast response |
| 11 | K1   | day  | 2.0     | 30     | Storage coefficient of fast response |
| 12 | L    | mm   | 0.0     | 100    | Reservoir threshold |
| 13 | PERC | mm d$^{-1}$ | 0.0 | 100 | Percolation rate |
| 14 | K2   | day  | 30      | 250    | Storage coefficient of slow response |
| 15 | BMAX | day  | 0.0     | 30     | Max baseflow of low flows |

**Table A2.** Selected parameters and their boundaries for the SAC model. The parameter boundaries are based on Newman et al. (2015), the Priestley-Taylor parameter has been adapted based on Lhomme (1997).

| | Name | Unit | Lower boundary | Upper boundary | Description |
|---|---|---|---|---|---|
| 1 | MFMAX | mm $°C^{-1}$ $6h^{-1}$ | 0.8 | 3.0 | Max snow melt factor |
| 2 | MFMIN | mm $°C^{-1}$ $6h^{-1}$ | 0.01 | 0.79 | Min snow melt factor |
| 3 | UADJ | km $6h^{-1}$ | 0.01 | 0.40 | Wind adjustment factor for rain on snow |
| 4 | SI | mm | 1.0 | 3500 | snow water equivalent for 100% snow area |
| 5 | SCF | - | 0.1 | 5.0 | Snow undercatch correction factor |
| 6 | PXTEMP | $°C$ | -3.0 | 3.0 | Temperature for rain/snow transition |
| 7 | PT | - | 1.0 | 1.74 | Priestley-Taylor coefficient |
| 8 | UZTWM | mm | 1.0 | 800 | Upper zone max storage of tension water |
| 9 | UZFWM | mm | 1.0 | 800 | Upper zone max storage of free water |
| 10 | UZK | $day^{-1}$ | 0.1 | 0.7 | Upper zone free water lateral depletion rate |
| 11 | ZPERC | - | 1.0 | 250 | Max percolation rate |
| 12 | REXP | - | 0.0 | 6.0 | Exponent of the percolation equation |
| 13 | PFREE | - | 0.0 | 1.0 | Frac. percolating from upper to lower zone |
| 14 | LZTWM | mm | 1.0 | 800 | Lower zone max storage of tension water |
| 15 | LZFPM | mm | 1.0 | 800 | Lower zone max storage of free water |
| 16 | LZFSM | mm | 1.0 | 1000 | Lower zone max storage of sec. free water |
| 17 | LZPK | $day^{-1}$ | $1^{-5}$ | 0.025 | Lower zone prim. free water depletion rate |
| 18 | LZSK | $day^{-1}$ | $1^{-3}$ | 0.25 | Lower zone sec. free water depletion rate |

**Table A3.** Selected parameters and their boundaries for the VIC model based on Demaria et al. (2007); Chaney et al. (2015); Melsen et al. (2016); Mendoza et al. (2015).

| | Name | Unit | LB | UB | Description |
|---|---|---|---|---|---|
| 1 | Tsmax | °C | 0.0 | 3.0 | Max temp. where snowfall can occur |
| 2 | Tsmin | °C | Tsmax-0.01 | Tsmax-3.0 | Min temp. where rainfall can occur |
| 3 | SR | - | $5 \cdot 10^{-5}$ | 0.5 | Surface roughness of the snow pack |
| 4 | NewAlb | - | 0.7 | 0.99 | New snow albedo |
| 5 | albaa | - | 0.88 | 0.99 | Base in snow albedo function for accum. |
| 6 | albtha | - | 0.66 | 0.98 | Base in snow albedo function for melt |
| 7 | RZT1 | - | 0.5 | 2 | Multipl. factor rootzone thickness layer 1 |
| 8 | RZT2 | - | 0.5 | 2 | Multipl. factor rootzone thickness layer 2 |
| 9 | RZT3 | - | 0.5 | 2 | Multipl. factor rootzone thickness layer 3 |
| 10 | Rmin | - | 0.1 | 10 | Multipl. factor min. stom. res. vegetation |
| 11 | Bi | - | $10^{-5}$ | 0.4 | Infiltration shape parameter |
| 12 | Depth2 | m | 0.1 | 3.0 | Depth of soil layer 2 |
| 13 | Expt2 | - | 4.0 | 30 | Exponent of the Brooks-Corey relation |
| 14 | Ds | - | $10^{-4}$ | 1.0 | Frac. Dsmax non-linear baseflow starts |
| 15 | Dsmax | mm d$^{-1}$ | 0.1 | 50 | Max velocity of the baseflow |
| 16 | Ws | - | 0.2 | 1.0 | Frac. Wsmax non-linear baseflow starts |
| 17 | Depth3 | m | 0.1 | 3.0 | Depth of soil layer 3 |

**Appendix: B. Change in sensitivity versus temperature and precipitation**

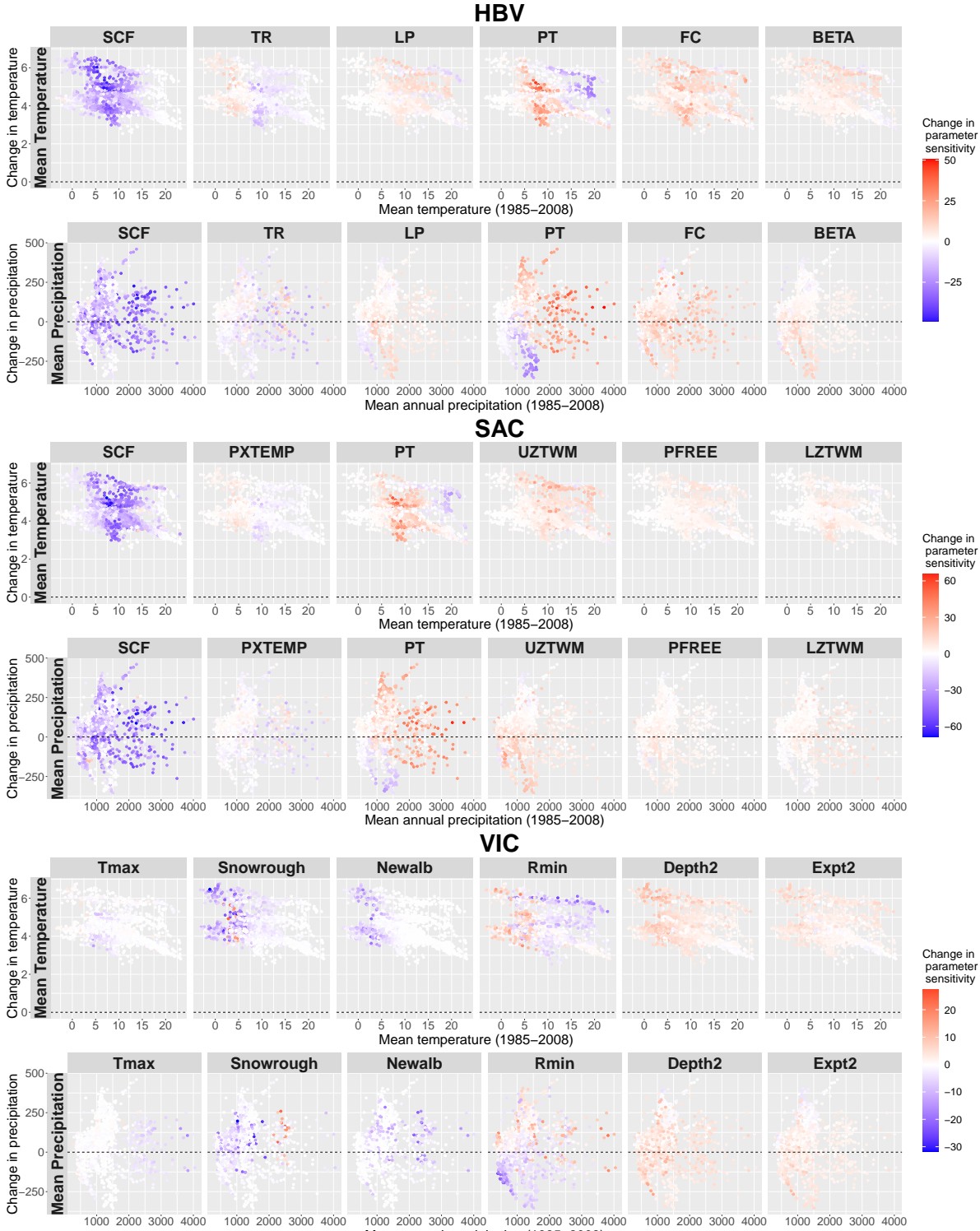

**Figure A1.** Change versus historical values in mean temperature and mean precipitation over 605 basins, with change in parameter sensitivity indicated using mean discharge as target variable. All three climate scenarios are shown together in each subplot. Parameter sensitivity for the historical period is expressed as dot size. Change in parameter sensitivity in colour. Red colours indicate an increase in sensitivity, blue a decrease.

**Appendix:  C. Results for discharge timing as target variable**

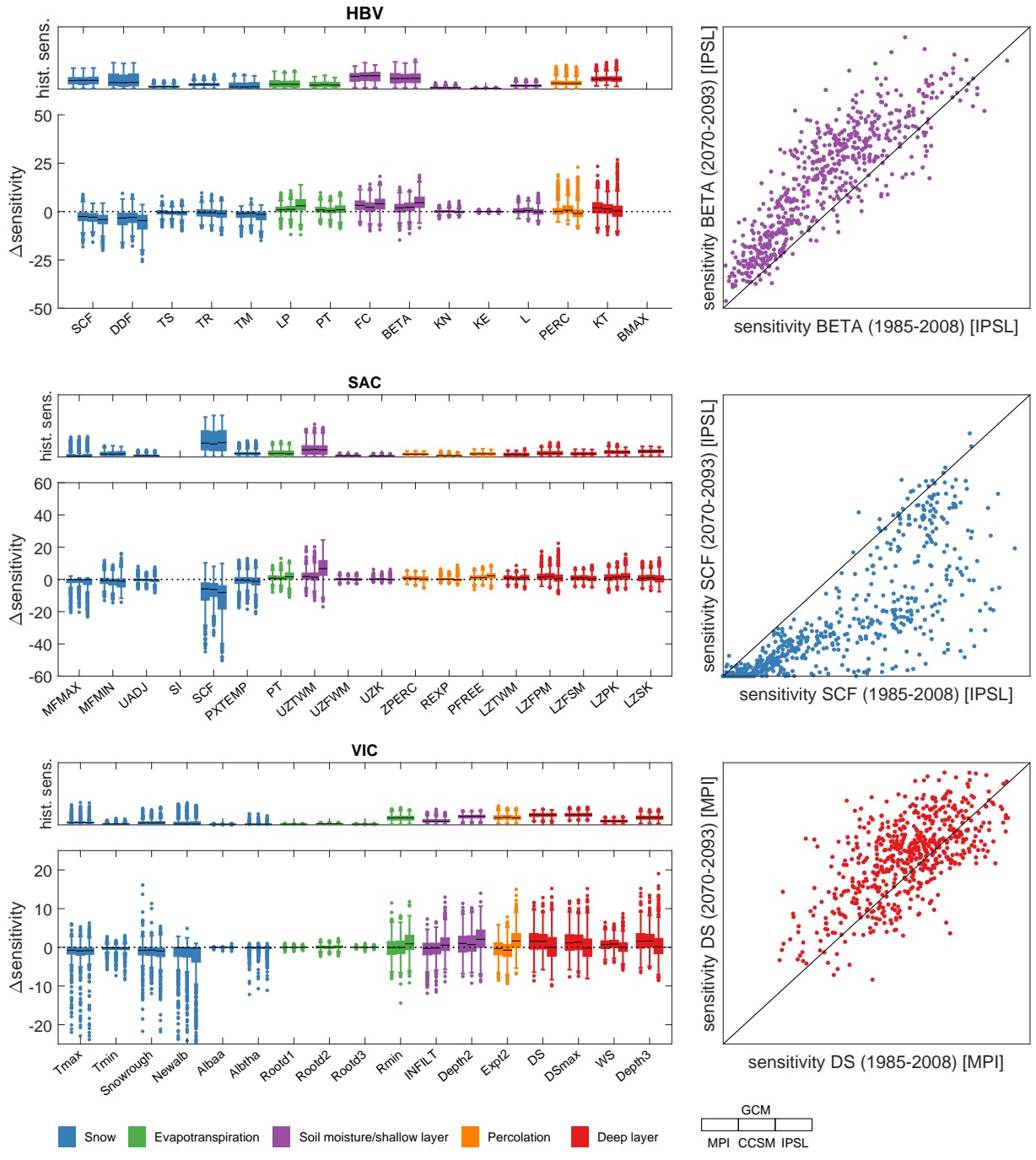

**Figure B1.** The distribution of change in parameter sensitivity (Δsensitivity) over 605 basins for the period 2070-2093 compared to 1985-2008, displayed per parameter per model, for three different GCM forcings for the discharge timing. Above each Δsensitivity panel, historical sensitivity is displayed. The panels on the right show the data for a selected case per model.

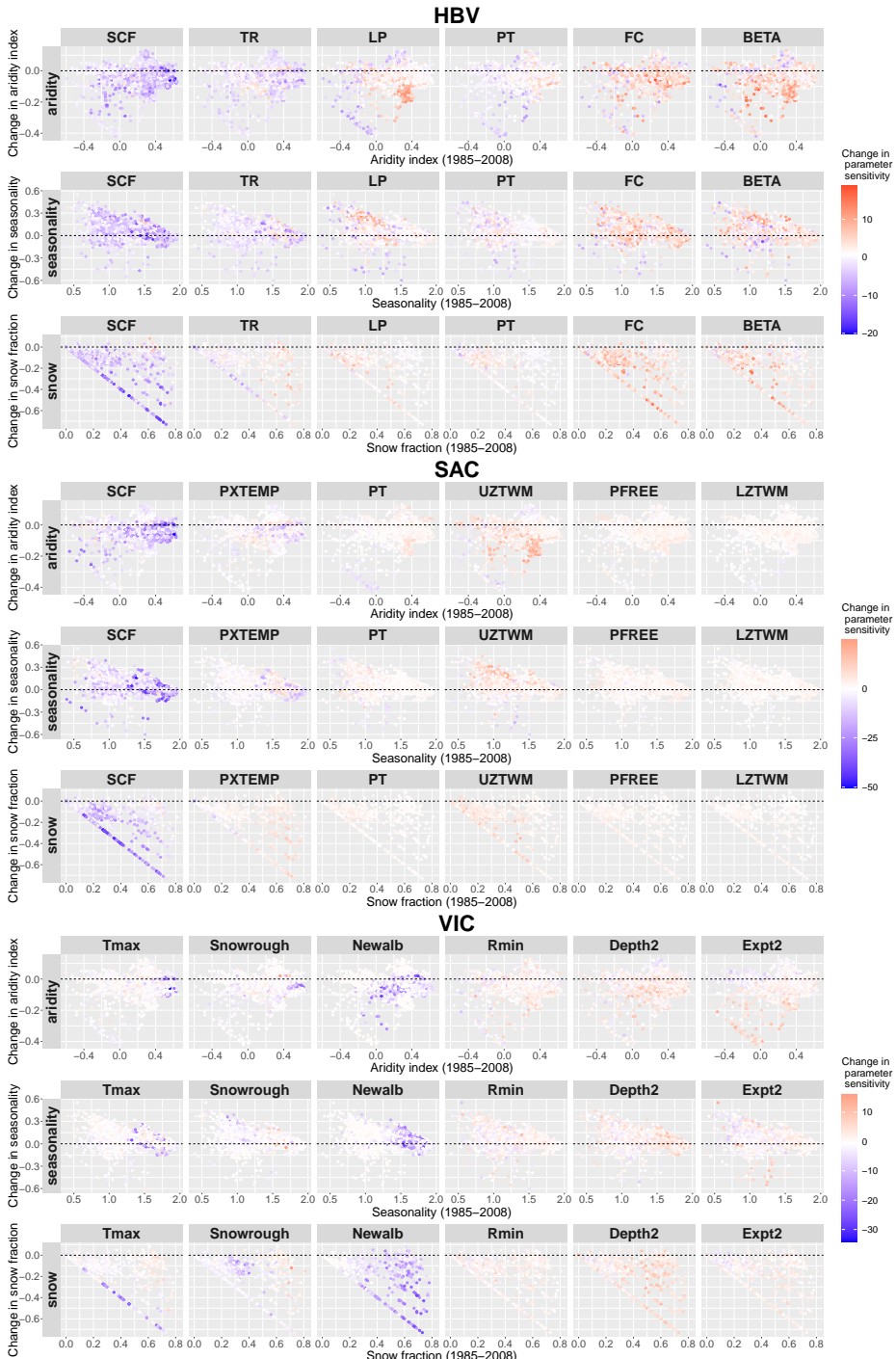

**Figure B2.** Change in parameter sensitivity versus historical climate indicators and change in climate indicators for 605 basins using discharge timing as target variable. All three climate scenarios are shown in the plots. The climate indicators are aridity index (-1 highly arid, +1 highly humid), seasonality, and fraction of precipitation falling as snow, as defined by Knoben et al. (2018). Parameter sensitivity for the historical period is expressed in dot size, change in parameter sensitivity in colour: red indicates an increase in sensitivity, blue a decrease.

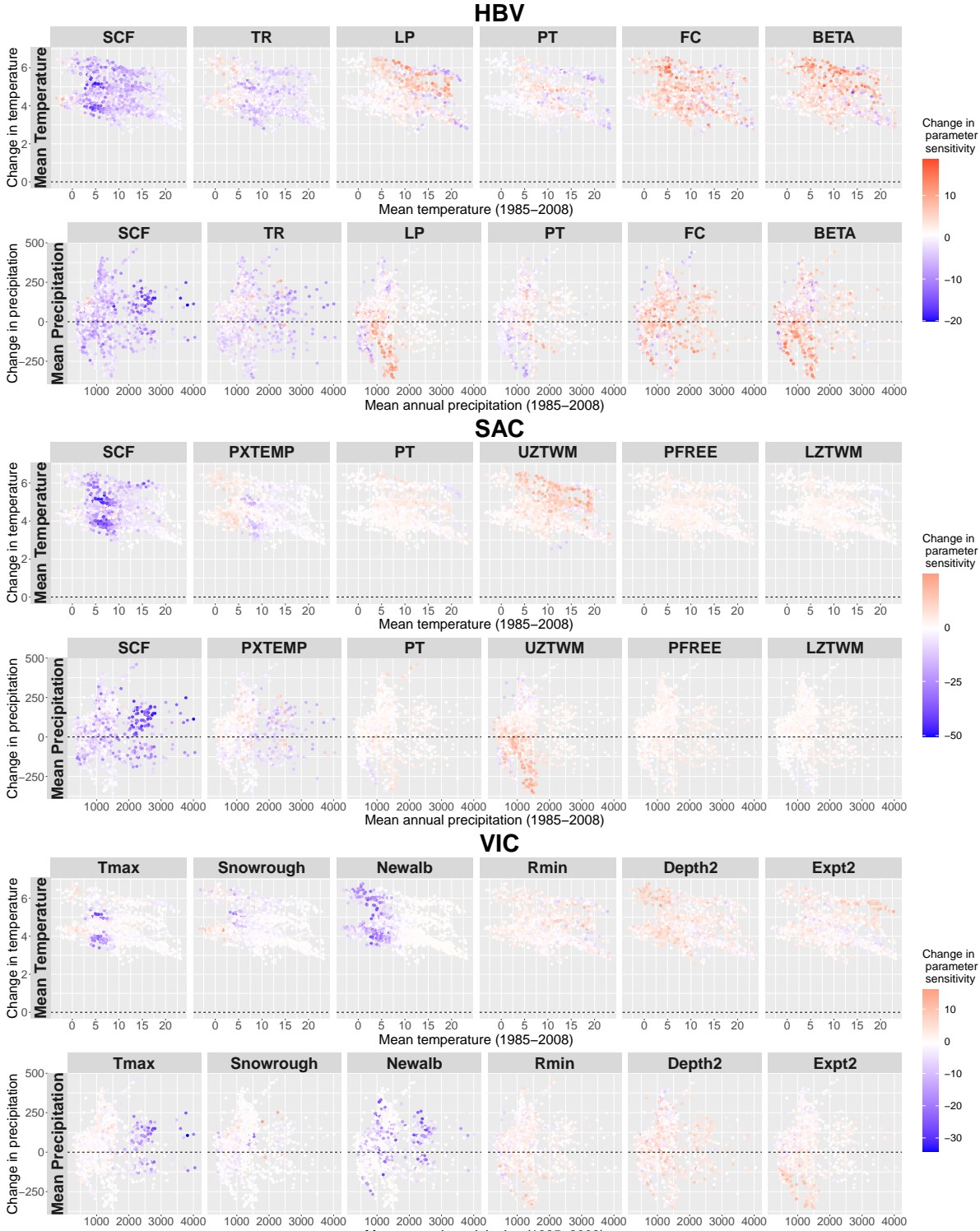

**Figure B3.** Change versus historical values in mean temperature and mean precipitation over 605 basins, with change in parameter sensitivity indicated using discharge timing as target variable. All three climate scenarios are shown together in each subplot. Parameter sensitivity for the historical period is expressed as dot size. Change in parameter sensitivity in colour. Red colours indicate an increase in sensitivity, blue a decrease.

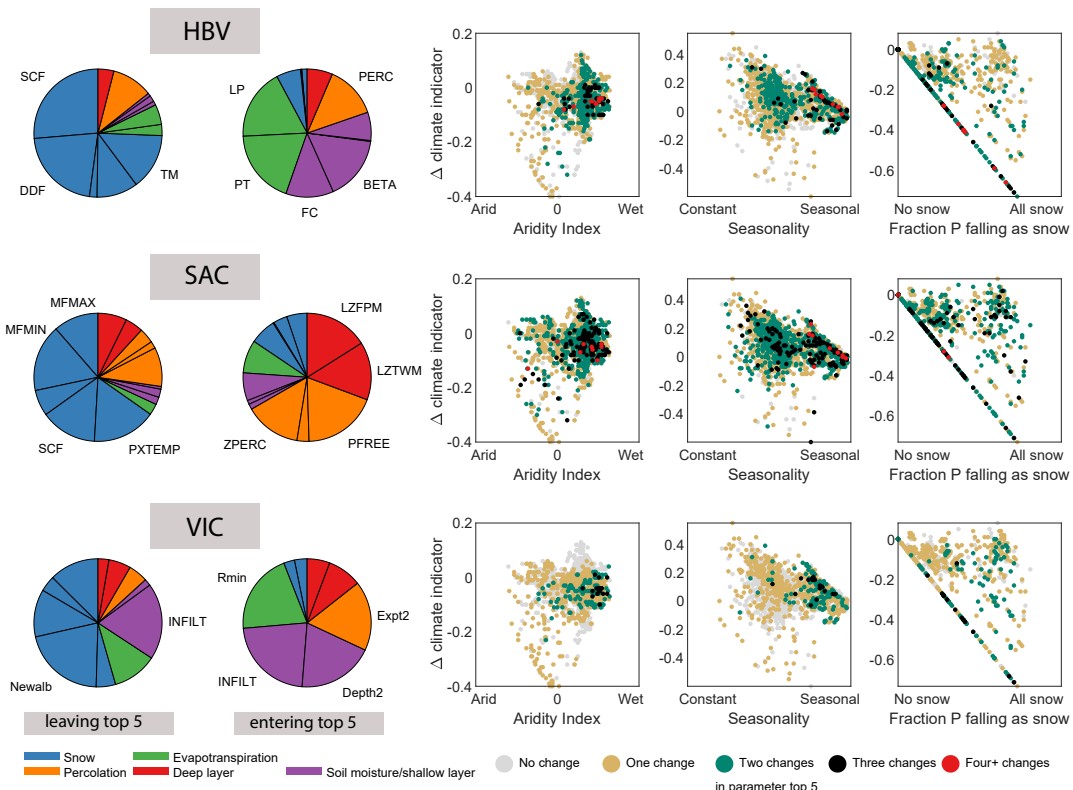

**Figure B4.** Impact of change in parameter sensitivity on top-5 position for the discharge timing, where top-5 refers to the five most sensitive parameters per basin - generally the parameters that are calibrated. The pie charts show which parameters leave the top-5 (left) and which parameters enter the top-5 (right). The right panels relates the number of changes in the parameter top-5 to climate and climate change indicators.

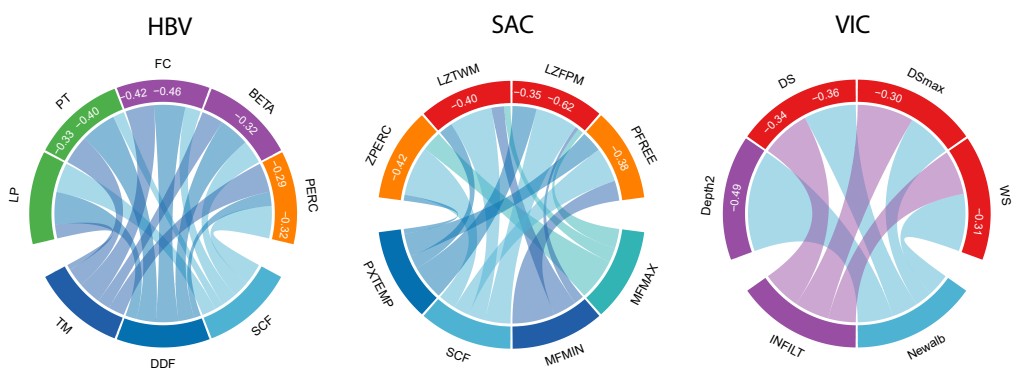

**Figure B5.** Indication of parameter sensitivity transmission for the discharge timing - the day of the year that half of the discharge has passed. The chord (circle) diagrams display transmission of sensitivity, indicated with a band from the parameter that decreases in sensitivity to the parameter that increases in sensitivity. The width of the band indicates the strength of the negative correlation. The example from panel a is indicated with an arrow in the chord diagram of HBV. The white number indicates the strength of the correlation. In all three chord diagrams, the lower part shows the parameters that decrease in sensitivity, and the upper part the parameters that increase in sensitivity, with the white number indicating the strength of the correlation (for clarity, negative correlations lower than 0.32 are not displayed). Colors are according to the process they represent (with different shades of blue used for snow parameters for clarity). The chord diagrams are focused around the most relevant parameters based on Fig. 6.

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
