# Peer review of "Climate change impacts model parameter sensitivity - implications for calibration strategy and model diagnostic evaluation"

_Hydrology and Earth System Sciences, 2020_

## Referee Comment (RC1) · Anonymous Referee #1 · 9 Jun 2020

This study investigates the changes in parameter sensitivity for a hydrological model under a plausible rate of climate change. This is considered in the context of model calibration, i.e. what would happen if one were to calibrate only the most sensitive parameters. This experiment is performed using the DELSA sensitivity method across 605 catchments in the U.S. with the SAC-SMA, VIC, and HBV models in a historical and future period forced by a GCM. Results show that some parameters, especially snow, show decreasing sensitivity, while others increase in unpredictable ways.

This is an interesting and novel research question that is addressed with a well-devised and executed experiment. The large sample of catchments and comparison of Knoben

indicators is very thorough. I fully support its publication, but I have some minor questions about the framing and interpretation of results.

1. The motivation related to calibration is somewhat unexpected. I am not sure how common is the practice of calibrating only the five most sensitive parameters. For these lumped catchment models, a calibration of 5, 15, or 30 parameters is computationally not very difficult, though there is the concern of equifinality.

It is probably not necessary because the paper would be just as interesting if framed as the change in parameter sensitivity over long timescales under climate change. The study does not perform a calibration, and does not consider how the calibrated values of the parameters would change due to climate. For example, Section 3.3 is not really considering the impact on model calibration, instead it is considering the impact of climate change on the ranking of sensitive parameters.

This is a minor clarification in a few places in the paper, but it is one possible point of improvement.

2. I imagine many readers will be interested in the diagnostic question: what can the sensitivity analysis tell us about hydrologic processes changing in the future? There are a few clear examples of this in the results, such as the decrease in snow processes, and the increase in ET processes. However, despite the very thorough experiment and comparison across climate indicators, there is not much relationship between the level of climate change and the change in parameter sensitivity across models.

The authors have a good discussion of what this could mean, that perhaps there is no consensus how the hydrological system will change in the future. My somewhat pessimistic interpretation was that the increases in parameter sensitivity do not follow any process-based reasoning, and are only the result of the simplified conceptual model structure. Additionally, it is not possible to say whether parameters are more sensitive because the processes are occurring more frequently, or with higher magnitude, or only because some other process is not occurring and the residual sensitivity had to

go somewhere.

There is nothing quantitative to do about this, but it is a very interesting issue. I would encourage the authors to consider focusing discussion more on this point, and perhaps a bit less on the calibration-related issues.

3. There is some opportunity to relate this study to previous studies of time-varying sensitivity on much shorter timescales (event or seasonal). In those cases, the temporal dynamics of sensitivity can be directly linked to flood or drought events. The change in parameter sensitivity here is expected, because of course the catchment is not stationary on a daily timestep. However in the climate change case, the driving processes are less clear, which raises more concerns about structural issues.

I am curious whether the authors view the current study as part of a continuum across timescales, or as a separate matter entirely.

—————————————————

---

## Referee Comment (RC2) · Anonymous Referee #2 · 12 Jun 2020

Hydrological models play a crucial role in the projection of future water resources and extremes including drought and high flows under climate change. Parameter calibration is key to whether models could produce reliable simulations. This study focuses on the change of parameter sensitivity based on discharge under climate change through ideal experiments over 605 basins in the U.S. and offers good guidance to modelers about parameter transferability under different climates. This work is novel and clearly organized. However, it still needs some revisions before publication.

General comments:

1. The introduction is too short and did not give a full review of the literature. The

authors could add some studies about climate change and its impacts on hydrology, especially in the U.S.. There are only several studies about how climate influences parameter sensitivity that are cited in this study.

2. In this study, the parameter range is defined as full, however, the range of parameters influence the parameter sensitivity analysis. I wonder whether the results are robust regardless of the selected ranges of parameters. Besides, whether the change of parameter sensitivity is related to catchment physical properties like catchment area, elevation, etc. (Saft et al., 2016)?

3. What is the change of precipitation, temperature in RCP8.5 over the selected 605 basins? A deeper analysis of the meteorological forcings is needed and would contribute to understanding the change of parameters and hydrological processes in models under climate change.

Specific comments: L10: The percentages of catchments with two parameter changes are quite small and negligible.

L35: There is lacking literature reviews about the hydrological parameters under different climates in the introduction. To my understanding, this work is quite relevant to some studies about the temporal transfer of parameters (Coron et al., 2012; Patil and Stieglitz, 2015; Shin et al., 2013).

L75: Why this study selected the output of CCSM? Only one GCM is selected in this work, however there are significant uncertainties in the outputs of GCMs and some studies used the ensemble to reduce the uncertainties. It is better to compare multiple outputs of GCMs.

L77: What is the specific bias correction method used in this study? And how did you select 605 from 671 catchments derived from CAMELS?

L118: "2.4 Analysis of sensitivity" is similar to "2.3 Sensitivity analysis". It is better to rename section 2.4.

L158: How meteorological fields are changed in RCP8.5 over the 605 basins is still unclear. It may be better to show the change of meteorological variables before sensitivity analysis.

L175: "there are parameters associated to all four processes besides snow", here you mean to exclude snow process? And you may change the words as ". . . expect snow"?

L182-L183: The conclusion is too harsh, as there is no clear correlation between AI and the change of sensitivity.

Figure 4: the labels of the X-axis are all climate indicators, it is better that you use AI, seasonality, and fraction of climate indicators.

Figure 6: The figures could be labeled as "(a), (b), . . ." and it is not easy to read correspondingly. The strong negative correlation is not quite obvious in Fig 6.

4 Discussion: There are discrepancies among the changes of parameter sensitivity based on HBV, SAC, and VIC. The authors could discuss how model structures affect parameter sensitivity.

Please also note the supplement to this comment:
https://www.hydrol-earth-syst-sci-discuss.net/hess-2020-179/hess-2020-179-RC2-supplement.pdf

---

## Author Comment (AC1) · 12 Jun 2020

We would like to thank the reviewer for the positive and constructive review. Here below we respond to the three points about framing and interpretation.

1) We are not entirely sure whether we agree with the reviewer on the unexpected motivation from calibration. Indeed computationally it is feasible to calibrate 30 parameters for simple models (as far as they are still simple with 30 parameters), but it is an ill-defined problem, for a given objective function only a limited number of parameters can be detected as dominant. Besides, the number of required model simulations to cover the entire parameter space in the same quality is increasing nonlinearly (see

[Figure]

Guse et al., 2020, Vol. 65, Issue 7, HSJ). Therefore, calibration can/will be more successful with parameter prioritization based on sensitivity analysis. On the other hand, we agree that it is easy to resolve the issue raised by the reviewer; the results are relevant independent of presenting it in the calibration context. We propose to wait for the opinion of the other reviewer on the framing, and then re-evaluate the framing.

2) It is indeed interesting (and alarming!) that the models disagree on process changes. We will further strengthen the discussion on this part.

3) This study indeed relates to earlier work on time-varying sensitivity. We make a very short link to this work (line 288 on page 14), but will further strengthen this discussion. We perceive our work as a continuum across timescales; basically we test for the average effect on the climate time scale of time varying sensitivity (and as such should indeed be representative of changing processes).

---

## Author Comment (AC2) · 13 Jun 2020

We would like to thank the reviewer for the constructive suggestions to improve our study. Below a point-by-point discussion in response to the review.

1) We chose for a short and concise introduction; the intro is therefore indeed not complete in terms of literature. We will expand the introduction with a section on hydrological climate change impacts in the US and a more extensive citation of studies that relate climate to sensitivity.

2) The parameter ranges might indeed impact the sensitivity analysis. Therefore, we

used the default ranges for each of the models, so that our study mimics applications of these models as good as possible.

Concerning the impact of catchment physical properties: since we conducted a global sensitivity analysis, the parameters have not been calibrated to the local situation. Only the VIC models contain land-surface information that is usually not calibrated, but we also applied sensitivity analysis to these terms (LAI and rooting depth through a multiplication factor). In HBV, the only catchment physical property that is (obviously) not included in the sensitivity analysis is the elevation, but the effect of elevation is conveyed through the forcing. As such, there might be a small effect of physical properties on sensitivity in VIC (because multiplication factor is applied to the the initial LAI and rooting depth values) but these parameters were found not to be highly sensitive anyways, and we don't expect any effect of physical properties for HBV and SAC, in this context of global sensitivity analysis.

3) This is a useful suggestion. We will add, before presenting the sensitivity results, a short summary of the projected temperature and precipitation changes for the selected basins.

L35: Yes, we will add more relevant literature on the relation between climate and sensitivity.

L75: We selected only one GCM as we see this study as a 'proof of concept', therefore we also talk about 'a plausible climate change rate' rather than an absolute projection. We will elaborate on our choice and also place CCSM in context of other GCM's.

L77: The applied bias correction methods is the bias-correction and spatial disaggregation (BCSD) method of Wood et al. (2004). We will add this to the text.

The selection of 605 catchments compared to the 671 that are available in CAMELS is because at the time of performing these calculations, the other 66 catchments still had some issues with catchment area (two datasets disagreed more than 10% on

catchment area, thereby influencing the spatial averaging of the forcing). We will clarify this in the text.

L118: Agreed, this can indeed cause confusion and will therefore be reformulated.

L158: Agreed, we will add this.

L175: Yes, we meant: 'on top of the snow process,..'. We will think of a way to better formulate this.

L182: It is not completely clear to us what the reviewer refers to here. In this sentence, we write: "From the figure, it can be seen that the patterns relating parameter sensitivity to climate and climate change indicators are weak. The aridity index seems to have most explanatory value, followed by seasonality and fraction precipitation falling as snow, respectively". Does the reviewer mean that the second sentence gives too much attention to a non-existing correlation? Our intention was to make that clear from the preceding sentence, but we will reformulate it.

Fig. 4: Agreed, will be adapted.

Fig. 6: Agreed, labels will be added. It is not completely clear to us what the reviewer means with that the negative correlation is not quite obvious. Does this refer to the left panel? Because the chord diagrams are only showing strong negative correlations (no positive correlations). We will try to reformulate the caption and can perhaps further clarify the figure.

Discussion: Yes, also in response to the other review, we will elaborate on the role of model structure on parameter sensitivity and change in parameter sensitivity in the discussion.

We would like to thank the reviewer for the useful literature suggestions.

---

## Referee Comment (RC3) · Anonymous Referee #3 · 15 Jun 2020

General comments

This study analyses changes in sensitivity of model parameters due to changes in climate projections. The sensitivity and its changes are evaluated by using 3 different models in large sample of catchments in U.S. (CAMELS dataset).

In general I agree with two previous reviews, i.e. study is potentially interesting, but a revision/extension is needed/suggested. The main critical comments are:

1) Introduction does not fully cover studies that evaluated changes/temporal stability/sensitivity of model parameters in (observed) varying climate conditions, as well as studies evaluating different sensitivity approaches in hydrological modelling (e.g. De-

vak, Dhanya, 2017). This can improve the formulation the current state of the art of the problem and the research gaps.

2) Methods are not described in a sufficient detail and rigorous way. It will be very interesting to see similarities and differences between the models, including differences in model inputs and calculation of different runoff generation processes (snow accumulation and melt, evapotranspiration, soil moisture changes, etc.).

3) I agree with reviewer #1 that there is a missed opportunity to expand the sensitivity analysis to seasonal and event scales. The selection of target variable (i.e. mean annual runoff) limits the significance and contribution of the study. The impact of expected climate change on hydrological processes is interesting mainly because of changes in seasonal and event-based characteristics. The setup and results of using selected target variables is to some extent obvious and technical (i.e. not related to changes in the main runoff generation processes). For example for HBV model. It is clear (and expected) that in catchments with snow influence it is the SCF parameter which is sensitive to annual runoff, because it is the only one model parameter which can increase/decrease the precipitation input to the model. This is not related to climate change, it is a technical feature of the model. All the processes simulating accumulation/melt/runoff generation and routing are practically insensitive to long-term annual runoff. Similarly for arid catchments, it is only parameter representing limit for potential evaporation which can somewhat change the overall water balance. Why to test the sensitivity of other model parameters? For the reader it will be interesting to see some strategy and research hypotheses which parameters and why are expected to be sensitive in relation to climate change. So, this is why I fully support the comment asking to expand the analyses and to use some other target variables representing seasonal of event based runoff characteristics.

4) I would like to support the comment of reviewer #2 to expand the evaluation of results and to assess "the role of model structure on parameter sensitivity and change in parameter sensitivity". This can be, in my opinion part of the results not just part

in the discussion. Comparison and more detailed evaluation of three different types of models will for sure improve the significance of the results.

References

Manjula Devak, C. T. Dhanya; Sensitivity analysis of hydrological models: review and way forward. Journal of Water and Climate Change 1 December 2017; 8 (4): 557–575. doi: https://doi.org/10.2166/wcc.2017.149

---

## Short Comment (SC1) · 22 Jun 2020

We would like to thank the reviewer for the suggestions to improve the manuscript. Below a point-by-point response.

1) Agreed, we will expand the literature in the introduction.

2) We will expand the description of the methodology. Furthermore, we will add a summary of the direct model output (the different states/fluxes) by means of a boxplot to provide more insights on model functioning.

3) We think we do not completely agree with the reviewer at this point, although the

suggestion can be read in two ways.

The goal of this study is to evaluate if within a plausible climate change rate, parameter sensitivity changes. Evaluating variations in sensitivity at the seasonal and event scale is therefore out of the scope of this study – for this we refer to the discussion.

The reviewer suggestion also be read as a suggestion to evaluate timing-metrics beyond the mean discharge within the climate change context. This would indeed be interesting and valuable, but since we consider this study a 'proof of concept' we limit ourselves to the most straight forward metric – mean discharge. The reviewer is correct that parameter sensitivity depends on the metric of interest – indeed SCF in HBV will logically have substantial influence on the water balance in snow-dominated catchments. That is for the sensitivity itself. However, the change in sensitivity can in this case most likely be assigned to climate change. We evaluated two 23-year periods, with only the climate changed.

Indeed, when evaluating other metrics, other parameters might appear sensitive or demonstrate different changes in sensitivity. We totally agree that this is an interesting research line. We also acknowledge that we do not stress enough yet, that these results are only valid for mean discharge, and that for other (e.g. timing or event-related) metrics results might differ - we will emphasise this better. The main conclusion remains valid though: when using models for climate change impact studies, care should be taken in the calibration.

4) Agree, we will add this to the manuscript. This aligns very well with the suggestion under point 2, for which we will add the boxplots demonstrating the fluxes and states of the different models.

---

## Author Response (AR1)

Rebuttal

hess-2020-179

Climate change impacts model parameter sensitivity – What does this mean for calibration?

We would like to thank the editor for organizing the review process. We received three review reports. All three reviewers acknowledge the relevance of the study, and find the methods appropriate. The main comment that we distilled from the reviews is that the reviewers would appreciate a more thorough diagnostic discussion of how and why the models differ in their results. Therefore, we have broadened the scope of the revised manuscript. Below we provide a point-by-point response (with our response indicated in *italic*) to the issues raised by the reviewers.

We hope that editor and the reviewers are satisfied with our revision of the manuscript.

Best regards,
Lieke Melsen
Björn Guse

**Reviewer 1**

This study investigates the changes in parameter sensitivity for a hydrological model under a plausible rate of climate change. This is considered in the context of model calibration, i.e. what would happen if one were to calibrate only the most sensitive parameters. This experiment is performed using the DELSA sensitivity method across 605 catchments in the U.S. with the SAC-SMA, VIC, and HBV models in a historical and future period forced by a GCM. Results show that some parameters, especially snow, show decreasing sensitivity, while others increase in unpredictable ways. This is an interesting and novel research question that is addressed with a well-devised and executed experiment. The large sample of catchments and comparison of Knoben indicators is very thorough. I fully support its publication, but I have some minor questions about the framing and interpretation of results.

*We would like to thank the reviewer for the support and acknowledging the novelty of the research question.*

1. The motivation related to calibration is somewhat unexpected. I am not sure how common is the practice of calibrating only the five most sensitive parameters. For these lumped catchment models, a calibration of 5, 15, or 30 parameters is computationally not very difficult, though there is the concern of equifinality. It is probably not necessary because the paper would be just as interesting if framed as the change in parameter sensitivity over long timescales under climate change. The study does not perform a calibration, and does not consider how the calibrated values of the parameters would change due to climate. For example, Section 3.3 is not really considering the impact on model calibration, instead it is considering the impact of climate change on the ranking of sensitive parameters. This is a minor clarification in a few places in the paper, but it is one possible point of improvement.

*We agree with the reviewer that we do not perform any calibration, and that the title and headings could therefore be misleading. We still believe the calibration-perspective is highly relevant in the context of our research question, but have made sure that the word 'calibration' is everywhere replaced with 'calibration strategy' or 'calibration procedure'. Furthermore, we have broadened the scope of the manuscript, with more attention for diagnostic model evaluation. This is also reflected in the new title.*

2. I imagine many readers will be interested in the diagnostic question: what can the sensitivity analysis tell us about hydrologic processes changing in the future? There are a few clear examples of this in the results, such as the decrease in snow processes, and the increase in ET processes. However, despite the very thorough experiment and comparison across climate indicators, there

is not much relationship between the level of climate change and the change in parameter sensitivity across models. The authors have a good discussion of what this could mean, that perhaps there is no consensus how the hydrological system will change in the future. My somewhat pessimistic interpretation was that the increases in parameter sensitivity do not follow any process-based reasoning, and are only the result of the simplified conceptual model structure. Additionally, it is not possible to say whether parameters are more sensitive because the processes are occurring more frequently, or with higher magnitude, or only because some other process is not occurring and the residual sensitivity had to go somewhere. There is nothing quantitative to do about this, but it is a very interesting issue. I would encourage the authors to consider focusing discussion more on this point, and perhaps a bit less on the calibration-related issues.

*We have taken this suggestion from the reviewer wholeheartedly and have broadened the scope of the manuscript to not only focus on calibration, but also on model diagnostic evaluation. This changed the tone and also puts more emphasis on the model disagreement.*

3. There is some opportunity to relate this study to previous studies of time-varying sensitivity on much shorter timescales (event or seasonal). In those cases, the temporal dynamics of sensitivity can be directly linked to flood or drought events. The change in parameter sensitivity here is expected, because of course the catchment is not stationary on a daily timestep. However in the climate change case, the driving processes are less clear, which raises more concerns about structural issues. I am curious whether the authors view the current study as part of a continuum across timescales, or as a separate matter entirely.

*As we have now further clarified in the introduction (l.48-56 in the new manuscript), we see the results of our sensitivity analysis as representative for a systemic change: the result of a summation of events that have become more or less frequent in a future climate.*

**Reviewer 2**

Hydrological models play a crucial role in the projection of future water resources and extremes including drought and high flows under climate change. Parameter calibration is key to whether models could produce reliable simulations. This study focuses on the change of parameter sensitivity based on discharge under climate change through ideal experiments over 605 basins in the U.S. and offers good guidance to modelers about parameter transferability under different climates. This work is novel and clearly organized. However, it still needs some revisions before publication.

*We would like to thank the reviewer for acknowledging the novelty and the constructive suggestions to improve our study.*

General comments:
1. The introduction is too short and did not give a full review of the literature. The authors could add some studies about climate change and its impacts on hydrology, especially in the U.S. There are only several studies about how climate influences parameter sensitivity that are cited in this study.

*Whereas we acknowledge that our introduction was rather concise, we are also fully aware that it will never be possible to be completely exhaustive in terms of literature. We have added references about climate change in the US, and about the relation between climate and parameter sensitivity.*

2. In this study, the parameter range is defined as full, however, the range of parameters influence the parameter sensitivity analysis. I wonder whether the results are robust regardless of the selected ranges of parameters. Besides, whether the change of parameter sensitivity is related to catchment physical properties like catchment area, elevation, etc. (Saft et al., 2016)?

*The parameter ranges might indeed impact the sensitivity analysis. Therefore, we used the default ranges for each of the models, so that our study mimics applications of these models as good as possible.*
*Concerning the impact of catchment physical properties: since we conducted a global sensitivity analysis, the parameters have not been calibrated to the local situation. Only the VIC models contain land-surface information that is usually not calibrated, but we also applied sensitivity analysis to these terms (LAI and rooting depth through a multiplication factor). In HBV, the only catchment physical property that is (obviously) not included in the sensitivity analysis is the elevation, but the effect of elevation is conveyed through the forcing. As such, there might be a small effect of physical properties on sensitivity in VIC (because multiplication factor is applied to the the initial LAI and rooting depth values) but these parameters were found not to be highly sensitive anyways, and we don't expect any effect of physical properties for HBV and SAC, in this con-*

*text of global sensitivity analysis.*

3. What is the change of precipitation, temperature in RCP8.5 over the selected 605 basins? A deeper analysis of the meteorological forcings is needed and would contribute to understanding the change of parameters and hydrological processes in models under climate change.

*This is a useful suggestion. We have added a boxplot that demonstrates the mean temperature and precipitation change in the future across the 605 basins. See Figure 3 of the updated manuscript.*

Specific comments:

L10: The percentages of catchments with two parameter changes are quite small and negligible.

*Yes, they are indeed small. We mention them nonetheless, to indicate that there are some cases where two parameters change.*

L35: There is lacking literature reviews about the hydrological parameters under different climates in the introduction. To my understanding, this work is quite relevant to some studies about the temporal transfer of parameters (Coron et al., 2012; Patil and Stieglitz, 2015; Shin et al., 2013).

*The literature suggested by the reviewer refers to studies that have evaluated parameter stability across time/space. Whether the parameter value itself changes is different from whether the sensitivity of the parameter changes (which is what we evaluated in this study). We refer to the non-stationarity of parameter values in the Discussion (l. 421 in new manuscript).*

L75: Why this study selected the output of CCSM? Only one GCM is selected in this work, however there are significant uncertainties in the outputs of GCMs and some studies used the ensemble to reduce the uncertainties. It is better to compare multiple outputs of GCMs.

*We selected only one GCM as we see this study as a 'proof of concept', therefore we also talk about 'a plausible climate change rate' rather than an absolute projection. Instead, we decided to put more effort in running three different hydrological models. We have clarified this on l. 95-99 in the new manuscript.*

L77: What is the specific bias correction method used in this study? And how did you select 605 from 671 catchments derived from CAMELS?

*The applied bias correction methods is the Bias Correction and Spatial Disaggregation (BCSD) method of Wood et al. (2004). The selection of 605 catchments*

*compared to the 671 that are available in CAMELS is because at the time of performing these calculations, the other 66 catchments still had some issues with catchment area (two datasets disagreed more than 10% on catchment area, thereby influencing the spatial averaging of the forcing). This has been clarified in the text on l. 93 and l. 103-104, respectively.*

L118: "2.4 Analysis of sensitivity" is similar to "2.3 Sensitivity analysis". It is better to rename section 2.4.

*We agree and have reformulated section 2.4.*

L158: How meteorological fields are changed in RCP8.5 over the 605 basins is still unclear. It may be better to show the change of meteorological variables before sensitivity analysis.

*We agree, this is now displayed in Fig. 3 and presented before the sensitvitiy analysis results.*

L175: "there are parameters associated to all four processes besides snow", here you mean to exclude snow process? And you may change the words as ". . . expect snow"?

*This sentence has been reformulated (see l. 232 of new manuscript).*

L182-L183: The conclusion is too harsh, as there is no clear correlation between AI and the change of sensitivity.

*We were not sure where the reviewer was referring to, but have added the word 'relatively' to relax the statement.*

Figure 4: the labels of the X-axis are all climate indicators, it is better that you use AI, seasonality, and fraction of climate indicators.

*The labels have been adapted. This is now Figure 5 in the new manuscript.*

Figure 6: The figures could be labeled as "(a), (b), . . ." and it is not easy to read correspondingly. The strong negative correlation is not quite obvious in Fig 6.

*The figure has been adapted to further clarify what is depicted. This is Figure 7 in the new manuscript.*

4 Discussion: There are discrepancies among the changes of parameter sensitivity based on HBV, SAC, and VIC. The authors could discuss how model structures affect parameter sensitivity.

*Yes, also in response to the other reviewer, we have elaborated on the role of model structure on parameter sensitivity and change in parameter sensitivity in the results and the discussion. This is now also better reflected in the title of the manuscript.*

**Reviewer 3**

This study analyses changes in sensitivity of model parameters due to changes in climate projections. The sensitivity and its changes are evaluated by using 3 different models in large sample of catchments in U.S. (CAMELS dataset). In general I agree with two previous reviews, i.e. study is potentially interesting, but a revision/extension is needed/suggested.

*We would like to thank the reviewer for the constructive feedback and acknowledging that hte study is potentially interesting.*

The main critical comments are:

1) Introduction does not fully cover studies that evaluated changes/temporal stability/sensitivity of model parameters in (observed) varying climate conditions, as well as studies evaluating different sensitivity approaches in hydrological modelling (e.g. Devak, Dhanya, 2017). This can improve the formulation the current state of the art of the problem and the research gaps.

*Whereas we acknowledge that our introduction was rather concise, we are also fully aware that it will never be possible to be completely exhaustive in terms of literature. We have added references about the relation between climate and parameter sensitivity and sensitivity analysis methods.*

2) Methods are not described in a sufficient detail and rigorous way. It will be very interesting to see similarities and differences between the models, including differences in model inputs and calculation of different runoff generation processes (snow accumulation and melt, evapotranspiration, soil moisture changes, etc.).

*We have expanded the description of the methodology. Furthermore, we have added a summary of direct model output (the different states/fluxes) by means of a boxplot to provide more insights on model functioning. This is Figure 3 in the new manuscript.*

3) I agree with reviewer #1 that there is a missed opportunity to expand the sensitivity analysis to seasonal and event scales. The selection of target variable (i.e. mean annual runoff) limits the significance and contribution of the study. The impact of expected climate change on hydrological processes is interesting mainly because of changes in seasonal and event-based characteristics. The setup and results of using selected target variables is to some extent obvious and technical (i.e. not related to changes in the main runoff generation processes). For example for HBV model. It is clear (and expected) that in catchments with snow influence it is the SCF parameter which is sensitive to annual runoff, because it is the only one model parameter which can increase/decrease the precipitation input to the model. This is not related to climate change,

it is a technical feature of the model. All the processes simulating accumulation/melt/runoff generation and routing are practically insensitive to long-term annual runoff. Similarly for arid catchments, it is only parameter representing limit for potential evaporation which can somewhat change the overall water balance. Why to test the sensitivity of other model parameters? For the reader it will be interesting to see some strategy and research hypotheses which parameters and why are expected to be sensitive in relation to climate change. So, this is why I fully support the comment asking to expand the analyses and to use some other target variables representing seasonal of event based runoff characteristics.

*The goal of this study is to evaluate if within a plausible climate change rate, parameter sensitivity changes. Evaluating variations in sensitivity at the seasonal and event scale is therefore out of the scope of this study – as we now explain in the introduction, we evaluate a longer period where the change in sensitivity would be the result of changes in certain types of events occurring more or less frequent. We refer to this as systemic change. This is a different approach from the event-based sensitivity analysis studies.*
*The reviewer suggestion can also be read as a suggestion to evaluate timing-metrics beyond the mean discharge within the climate change context. This would indeed be interesting and valuable, but since we consider this study as a 'proof of concept' we limit ourselves to the most straight forward metric – mean discharge. The reviewer is correct that parameter sensitivity depends on the metric of interest – indeed SCF in HBV will logically have substantial influence on the water balance in snow-dominated catchments. That is for the sensitivity itself. However, the change in sensitivity can in this case most likely be assigned to climate change. We evaluated two 23-year periods, with only the climate changed. Indeed, when evaluating other metrics, other parameters might appear sensitive or demonstrate different changes in sensitivity. In the discussion, we have put more emphasize on the fact that our results are only valid for mean discharge as target variable, see Section 4.1 in the new manuscript.*

4) I would like to support the comment of reviewer #2 to expand the evaluation of results and to assess "the role of model structure on parameter sensitivity and change in parameter sensitivity". This can be, in my opinion part of the results not just part in the discussion. Comparison and more detailed evaluation of three different types of models will for sure improve the significance of the results.

*We agree on this point and have therefore broadened the general scope of the manuscript.*

[revised manuscript text omitted]

---

## Author Response (AR2)

Rebuttal

hess-2020-179

Climate change impacts model parameter sensitivity - implications
for calibration strategy and model diagnostic evaluation

We would like to thank the editor for organizing the review process, and the re-
viewers for evaluating our re-submission. From the reviews that we received, and
the editor's interpretation of these reviews, we receive the signal that we might
have given the impression that we did not substantially address the reviews
in the previous round. This was by no means our intention; we substantially
altered the text and added and edited several figures. But it is clear that we
missed the signal in the previous reviews that extra calculations were requested,
and we apologize for missing this signal.

For this re-submission, we conducted many extra calculations. We would like to
stress that the study was already conducted based on 3 hydrological models and
605 catchments, which is already more comprehensive than many other studies,
especially given that we conduct sensitivity analysis which is computationally
intensive. For our previous submission, we conducted (1600+1800+1900)*605
= 3,206,500 (so more than 3 million) model runs. Therefore, there is always a
choice in which dimensions are relevant, and need to be explored in more de-
tail. The evaluation of the reviewers is that more signatures and more forcing
datasets need to be explored. We have expanded our analysis with two extra
forcing-datasets requiring more than 6 million extra model runs and redid all
the analysis for a second signature. We hope that this shows that we are willing
to seriously account for the evaluation of the reviewers. Increasing the dimen-
sions of evaluating does mean, however, that again the structure had to change
and that we had to make choices in what to show and present, and what not.

Best regards,

Lieke Melsen
Björn Guse

**Reviewer 1**
Accepted as is.

**Reviewer 2**
I want to thank the authors for considering at least part of my comments and suggestions. The clarity of presentation was improved; however, from a contribution in HESS I would still expect some more significant novel contribution. The limits of this study and most of the interpretations made are:

We would like to thank the reviewer for the feedback. As we explained in our response to the editor on the previous page, we had the impression that this study already was more comprehensive that many other studies in HESS considering that we explored three models in 605 basins, for which we conducted global sensitivity analysis. But since we received the same signal from the other reviewer, it is clear that we needed to expand the analysis.

(a) the use of only one climate scenario which is rather extreme in terms of an increase in air temperature and not significant (and mostly only increase) precipitation change;

We have now done simulations with two additional climate models. These models were selected to capture spread in the GCMs, and as such we more extensively sample plausible climate change space. Unfortunately, we had no bias corrected GCM runs with lower emission scenario's at our disposal, but by adding the two other GCMs we already capture a broader range in changes in temperature and precipitation.

(b) the evaluation is focusing on annual runoff only. The use of a large number of catchments with such setting (one extreme scenario, annual runoff, using only simulations not observations) does not show enough variability which can shed some more light on the differences between the models, so the interpretations and findings are either expected (obvious) or read more as speculation (are not supported by some evidence) or applies only to the very specific setting of the experiment, and the generality of findings (for other regions, other scenarios, other metrics or models) is thus relatively low. At least evaluation of changes in seasonal signatures (and/or some less extreme climate scenario) will likely help to support some interpretations made.

We redid the analysis for a second signature: the day of the year that half of the discharge has passed. Indeed, the results are very much dependent on the signature selected, so this really added to our results and conclusion. We hope we now have adequately addressed the feedback of the reviewer.

**Reviewer 3**

I would like to thank the authors for addressing my concerns about the relation between parameter sensitivity and climate change. However, I do not believe these comments were satisfactorily addressed in the revised manuscript. The main issue is the uncertainty caused by different forcing datasets and the selected parameter ranges, which is still unclear to me.

1. As mentioned in the revised manuscript, CCSM4 performs better in simulating precipitation and temperature. I still wonder how representative CCSM4 is in all projected climate of RCP8.5? Whether the different GCM outputs could cause an impact on the results?

Our idea was that by plotting the results about sensitivity against climate change indicators, we could draw more general conclusions about how climate change explains changes in parameter sensitivity. However, in response to this feedback and the feedback from the other reviewer, we added simulations from two more GCMs. These GCMs were selected to represent different climate model families as can be found in the climate genealogy of Knutti et al. (2013). In this way we capture the variation among different GCMs, and more comprehensively sample plausible climate change space.

2. This study used full parameter ranges in global sensitivity analysis, but the selected ranges may influence the parameter sensitivities. The authors could address the impact of different ranges of the selected parameters on the parameter sensitivity.

Indeed the results of the global sensitivity analysis depend on the ranges chosen for the parameters. We did, however, completely follow the default ranges that are provided in the literature for the employed models, and therefore we think that we did what is most sensible in this respect.

3. As the parameter sensitivity is rarely connected with climate indicators, why not put fig.5 and Session 3.3 into the discussion rather than the key results, as we could get limited information from fig.5.

This is not possible because the CAMELS catchments are not a representative sample of climates and climate changes. As such, Section 3.2 should not be over-interpreted but merely be seen as some first idea of the results. Only when the catchments are clustered, e.g. based on the climate indicators, more general conclusions could potentially be drawn. Therefore, without Section 3.3, Section 3.2 also loses its validity, even though Section 3.3 did not provide unexpected or very clear results.

[revised manuscript text omitted]

---

## Author Response (AR3)

Rebuttal

hess-2020-179

Climate change impacts model parameter sensitivity - implications
for calibration strategy and model diagnostic evaluation

Dear editor,

Thank you for organising the review process. Below we respond to the minor
suggestions from the reviewers.

Best regards,

Lieke Melsen
Björn Guse

**Reviewer 2**

I want to thank the authors for the revisions made. In my opinion, the results now show more robust and general outcomes. I have only two last suggestions before recommending the manuscript for publication:

Please add some more details to the description of the methods, particularly how is the model used/setup. There is not a sufficient detail of information allowing reproducibility of work and the interpretation of some results in its current form. For example, what are the model inputs and parametrisation variant used for the VIC model? (is the VIC model controlled only by air temperature and precipitation changes?) Or how is potential evaporation estimated for HBV model? Why are results (increase in air temperature but a smaller change in runoff, evaporation, soil moisture, snow) of the VIC model different to the other two models?

It will be interesting to add some more details about the characteristics of catchments describing some results (i.e. extending description of some groups of catchments mentioned). For example are there some consistent patterns in catchment characteristics for catchments which have a change in sensitivity depending on the hydrologic model (l.11, l.13, l.335-337, etc.).

We extended the description of the model set-up in response to the first points raised by the reviewer, see line 93-96. The results of VIC deviate from the HBV and SAC models because this model has a completely different structure - HBV and SAC are more comparable in terms of structure.

Concerning the second point of the reviewer, linking to catchment properties; we believe that this analysis is not appropriate for the global sensitivity analysis that we conducted. Since we did a global sensitivity analysis, the models were not calibrated for specific catchments. This means that catchment properties are not reflected in the results - only climate properties. That is why we differentiated the results across climate, but not across catchment properties.

**Reviewer 3**

Thank the authors for addressing my concerns and the study has much been improved. Specific comments: Fig.5 is difficult to read.

The labels of Figure 5 have been increased and the legend has been reworked. Unnecessary legends have been removed.